# The Challenges of the Nonlinear Regime for Physics-Informed Neural Networks

**Andrea Bonfanti**
BMW AG, Digital Campus Munich
Basque Center for Applied Mathematics
University of the Basque Country
abonfanti001@ikasle.ehu.eus

**Giuseppe Bruno**
BMW AG, Digital Campus Munich
Giuseppe.GB.Bruno@bmw.de

**Cristina Cipriani**
Technical University of Munich
Munich Center for Machine Learning
Munich Data Science Institute
cristina.cipriani@ma.tum.de

## Abstract

The Neural Tangent Kernel (NTK) viewpoint is widely employed to analyze the training dynamics of overparameterized Physics-Informed Neural Networks (PINNs). However, unlike the case of linear Partial Differential Equations (PDEs), we show how the NTK perspective falls short in the nonlinear scenario. Specifically, we establish that the NTK yields a random matrix at initialization that is not constant during training, contrary to conventional belief. Another significant difference from the linear regime is that, even in the idealistic infinite-width limit, the Hessian does not vanish and hence it cannot be disregarded during training. This motivates the adoption of second-order optimization methods. We explore the convergence guarantees of such methods in both linear and nonlinear cases, addressing challenges such as spectral bias and slow convergence. Every theoretical result is supported by numerical examples with both linear and nonlinear PDEs, and we highlight the benefits of second-order methods in benchmark test cases.

## 1 Introduction

PINNs have became ubiquitous in the scientific research community as a meshless and practical alternative tool for solving PDEs. The first attempts to exploit machine learning models for PDE solutions can be traced back to two articles from the 90s [3, 20], while the model acquired its name and popularity through a later publication [31]. Due to the flexible structure of the architecture, PINNs can be used for forward and inverse problems [42] and efficiently exploited for more complex engineering practice such as constrained shape and topology optimization, and surrogate modeling [35, 16]. However, the usability of PINNs for such applications is often hindered by their slow training and occasional failure to converge to acceptable solutions. Due to the black-box nature of PINNs, it is challenging to analyze their training dynamics and convergence properties mathematically [19]. Nonetheless, rapid training and reliable convergence are crucial aspects of any PDE solver intended for engineering applications.

**Related works.** In this context, the NTK [15] viewpoint has yielded intriguing insights, particularly in the realm of linear PDEs [40]. Although based on the assumption of overparameterized networks, this perspective has proven valuable in highlighting various intrinsic pathologies in PINN training,

38th Conference on Neural Information Processing Systems (NeurIPS 2024).

such as spectral bias [39, 2, 29], the complexity of the loss landscape generated by the PDE residuals [19] and the nuanced interplay among components of the loss function [38]. The salient characteristics of the NTK in the infinite-width limit are the fact that is deterministic at initialization, constant during training, and it linearizes the training dynamics due to the sparsity of the Hessian of PDE residuals [22, 23].

**Our contributions.**    In this paper, we delineate the profound theoretical distinctions between the application of PINNs to linear versus nonlinear PDEs, elucidating the differences in their NTK behavior. We show that, even under the idealistic assumption of the infinite-width limit, the NTK framework fails in the nonlinear domain. Our novel contribution lies in demonstrating that the NTK is stochastic at initialization, it is dynamic during training, and is accompanied by a non-vanishing Hessian. Given the evolution of the Hessian throughout training, we emphasize the need of employing second-order methods for nonlinear PDEs. Furthermore, we analyze their convergence guarantees, revealing that even in linear scenarios, the utilization of second-order methods proves advantageous in mitigating the issue of spectral bias. As a second-order method, we employ Levenberg-Marquardt algorithm, a stabilized version of the well-known Gauss-Newton algorithm, which approximates the Hessian to make it computationally feasible even for large networks. It is important to note that our goal is not to propose a novel training algorithm but to demonstrate the benefits of using *any* second-order method. The reason is twofold: in the nonlinear regime, we achieve faster and better convergence, while in the linear regime, where fast convergence can be achieved by first-order methods, the advantage of second-order methods lies in their ability to alleviate spectral bias.

Our work is organized as follows: Section 2 introduces PINNs, and Section 3 covers the NTK theory, comparing its dynamics in linear and nonlinear PDEs. Section 4 examines the convergence guarantees of second-order optimization methods. Finally, Section 5 presents numerical experiments that validate our theoretical insights.

## 2   Physics-Informed Neural Networks

We address the following PDE formulated on a bounded domain $\Omega \subset \mathbb{R}^{d_{\text{in}}}$,

$$\begin{aligned} \mathcal{R}u(x) = f(x), \quad & x \in \Omega, \\ u(x) = g(x), \quad & x \in \partial\Omega. \end{aligned} \tag{1}$$

Here, the PDE is defined with respect to the differential operator $\mathcal{R}$, while the boundary and initial conditions are collected in the function $g$. Notice that $\Omega$ can be either a spatial or spatio-temporal domain, depending on whether the PDE is time-dependent or not. PINNs aim to approximate the PDE solution $u : \Omega \to \mathbb{R}^{d_{\text{out}}}$ with a neural network $u_\theta$ parametrized by $\theta$, which is a vector containing all the parameters of the network. The "Physics-Informed" nature of the neural network $u_\theta$ lies in the choice of the loss function employed for training

$$\mathcal{L}(\theta) = \frac{1}{2} \int_\Omega |\mathcal{R}u_\theta(x) - f(x)|^2 dx + \frac{1}{2} \int_{\partial\Omega} |u_\theta(x) - g(x)|^2 d\sigma(x),$$

where $\sigma$ denotes a measure on the surface $\partial\Omega$. In this work, we specifically focus on scenarios where the PDE involves a nonlinear differential operator. Moreover, without loss of generality we consider the case where $f(x) = 0$. Since the function $f(x)$ does not depend on the parametrization, all of our results hold also for the case when it is nonzero. Moreover, we express (1) as

$$\begin{aligned} R(\Phi[u](x)) = 0, \quad & x \in \Omega, \\ u(x) = g(x), \quad & x \in \partial\Omega, \end{aligned} \tag{2}$$

where $\Phi[u] : \mathbb{R}^{d_{\text{in}}} \to \mathbb{R}^{k \times d_{\text{out}}}$, defined as

$$\Phi[u](x) = [u(x), \partial_x u(x), \partial_x^2 u(x), \dots, \partial_x^k u(x)], \tag{3}$$

denotes a vector encompassing all (possibly mixed) derivatives of $u$ until order $k$, while $R : \mathbb{R}^{k \times d_{\text{out}}} \to \mathbb{R}$ represents a differentiable function of the components of $\Phi[u]$.

**Remark 2.1.** *The importance of the function $R$ lies in its ability to completely encode the nonlinearity of the PDE, while the term $\Phi$ remains linear. Furthermore, for numerous well-known nonlinear PDEs (such as Burgers' or Navier-Stokes equations), the function $R$ exhibits a distinctive structure as it takes the form of a second-order polynomial.*

To illustrate this, we consider the example of the inviscid Burgers' equation, which for $(\tau, x) \in \Omega$ is expressed as $\partial_\tau u + u \partial_x u = 0$, where $\tau$ represents time and $x$ the space variable. It follows that

$$\Phi[u](\tau, x) = [u(\tau, x), \partial_\tau u(\tau, x), \partial_x u(\tau, x)],$$
$$R(z_1, z_2, z_3) = z_2 + z_1 z_3.$$

## 3  Neural Tangent Kernel for PINNs

We now introduce and develop the NTK for PINNs, inspired by the definition in [40]. We employ a fully-connected neural network featuring a single hidden layer, as follows

$$u_\theta(x) := \frac{1}{\sqrt{m}} W^1 \cdot \sigma(W^0 x + b^0) + b^1, \tag{4}$$

for any $x \in \mathbb{R}^{d_{\text{in}}}$. Here, $W^0 \in \mathbb{R}^{m \times d_{\text{in}}}$ and $b^0 \in \mathbb{R}^m$ denote the weights matrix and bias vector of the hidden layer, while $W^1 \in \mathbb{R}^{d_{\text{out}} \times m}$ and $b^1 \in \mathbb{R}^{d_{\text{out}}}$ are the corresponding parameters of the outer layer. Additionally, $\sigma : \mathbb{R} \to \mathbb{R}$ is a smooth coordinate-wise activation function, such as the hyperbolic tangent, which is a common choice for PINNs. Furthermore, we adopt the NTK rescaling $\frac{1}{\sqrt{m}}$ to adhere to the methodology introduced in the original work [15]. This is crucial for achieving a consistent asymptotic behavior of neural networks as the width of the hidden layer approaches infinity. In the following, for brevity, we denote with $\theta$ the collection of all the trainable parameters of the network, i.e. $W^1, W^0, b^1, b^0$.

**Remark 3.1.** *For the sake of brevity, we focus on the case of neural networks with a single hidden layer. However, the outcomes derived in this scenario may be extended to deep networks. We leave this extension to future works and refer to [33, 34] for results on finite networks with multiple hidden layers.*

We consider the discrete loss on the collocation points $x_i^r \in \Omega$ and the boundary points $x_i^b \in \partial\Omega$,

$$L(\theta) = \frac{1}{2N_r} \sum_{i=1}^{N_r} |r_\theta(x_i^r)|^2 + \frac{1}{2N_b} \sum_{i=1}^{N_b} |u_\theta(x_i^b) - g(x_i^b)|^2, \tag{5}$$

where $r_\theta(x_i^r) = R(\Phi[u_\theta](x_i^r))$ indicates the residual term. Furthermore, $N_r$ and $N_b$ denote the batch size of, respectively, the collection of $\mathbf{x}^r = \{x_i^r\}_{i=1}^{N_r}$ and $\mathbf{x}^b = \{x_i^b\}_{i=1}^{N_b}$, which are the discrete data used for training. We now consider the minimization of (5) as the gradient flow

$$\partial_t \theta(t) = -\nabla L(\theta(t)). \tag{6}$$

Using the following notation

$$u_\theta(\mathbf{x}^b) = \left\{ u_{\theta(t)}(x_i^b) \right\}_{i=1}^{N_b}, \qquad r_\theta(\mathbf{x}^r) = \left\{ r_{\theta(t)}(x_i^r) \right\}_{i=1}^{N_r}, \tag{7}$$

we can characterize how these quantities evolve during the gradient flow, through the NTK perspective.

**Lemma 3.2.** *Given the data (7) and the gradient flow (6), then $u_\theta$ and $r_\theta$ satisfy the following*

$$\begin{bmatrix} \partial_t u_{\theta(t)}(\mathbf{x}^b) \\ \partial_t r_{\theta(t)}(\mathbf{x}^r) \end{bmatrix} = -K(t) \begin{bmatrix} u_{\theta(t)}(\mathbf{x}^b) - g(\mathbf{x}^b) \\ r_{\theta(t)}(\mathbf{x}^r) \end{bmatrix}, \tag{8}$$

*where $K(t) = J(t)J(t)^T$ and*

$$J(t) = \begin{bmatrix} \partial_\theta u_{\theta(t)}(\mathbf{x}^b) \\ \partial_\theta r_{\theta(t)}(\mathbf{x}^r) \end{bmatrix}. \tag{9}$$

*Proof.* The proof is presented in [40]. □

We provide more details about the construction of $J(t)$ in Appendix A. The matrix $K$ is also referred to as Gram matrix. The analysis of Gram matrices and their behavior in the infinite-width limit [4, 5] yields results akin to the NTK analysis. It is important to note that Lemma 3.2 is applicable to any type of sufficiently regular differential operator.

## 3.1 The difference between linear and nonlinear PDEs

In the work [40], PINNs have been thoroughly investigated using the NTK, but only in the case of linear PDEs. Additionally, [22] extensively explores the similar case of standard neural networks with linear output. In particular, they show that in the infinite-width limit, the NTK is deterministic under proper random initialization and stays constant during training. Thereby, the dynamics in (8) is equivalent to kernel regression and has an analytical solution expressed in terms of the kernel. As noted in [22], the constancy of the NTK during training is equivalent to the linearity of the model. This characteristic is related to the vanishing of the (norm of the) Hessian of the network's output in the infinite-width limit. These well-known results are reported in Appendix B. In [43], the same convergence results for Gram matrices hold for nonlinear PDEs when using networks as in (4) with a scaling of $\frac{1}{m^s}$, where $s > \frac{1}{2}$. However, this scaling is inconsistent with the NTK model, so we focus on the unexplored case where $s = \frac{1}{2}$. The novel contribution of our paper lies in demonstrating that in this regime this phenomenon does not hold true when dealing with nonlinear PDEs, which we prove in this section. The network architecture and its associated assumptions are relatively standard, so we refer to Assumption B.2 in Appendix B. However, it is essential to delineate the specific assumptions related to the nonlinear PDE.

**Assumption 3.3** (on $\mathcal{R}$). *The differential operator $\mathcal{R}$ is nonlinear, hence the function $R$ is nonlinear. Moreover, the gradient $\nabla R$ is continuous.*

The first distinction with linear PDEs arises in the convergence as $m \to \infty$ of the NTK at initialization.

**Theorem 3.4.** *Consider a fully-connected neural network given by* (4) *satisfying Assumption B.2. Moreover, the PDE satisfies Assumption 3.3. Then, under a Gaussian random initialization $\theta(0)$, it holds*

$$K(0) \xrightarrow{\mathcal{D}} \bar{K} \quad as\ m \to \infty,$$

*where the limit is in distribution and $\bar{K}$ is not deterministic, but its law can be explicitly characterized.*

*Proof.* A detailed proof is in Appendix C. However, the basic idea is to reformulate the kernel as

$$K(0) = \Lambda_R(0)\, K_\Phi(0)\, \Lambda_R(0)^T,$$

where the matrix $K_\Phi(0)$ enclose the linear components of $\mathcal{R}$, hence the derivatives of the network's output, while the matrix $\Lambda_R(0)$ depends on the gradient of $R$ (so its contribution is relevant just in the nonlinear case). We can establish the convergence in probability of $K_\Phi(0)$ to a deterministic matrix by taking advantage of the linearity of the operator $\Phi$ and commuting $\Phi$ and $\partial_\theta$ (see Lemma C.2). The matrix $\Lambda_R(0)$ only converges in distribution, since it is a function of the network output and its derivatives, whose limits are Gaussian Processes at initialization by Proposition C.1. □

Next, we focus on the NTK behavior during training.

**Proposition 3.5.** *Under Assumption B.2 on the network, and Assumption 3.3 on the PDE, assume additionally that $R$ is a real analytic function. Let $u_*$ be a solution of the corresponding PDE and suppose that for every $m \in \mathbb{N}$ there exists $t_m$ such that*

$$\|u_\theta(t_m) - u_*\|_{\mathcal{C}^k} \le \varepsilon_m, \ \ with\ \varepsilon_m \to 0\ as\ m \to \infty. \tag{10}$$

*Finally, let $\theta(t)$ be obtained through gradient flow as defined in* (6) *and denote by $K(t)$ the corresponding NTK. For $\theta(0) \sim \mathcal{N}(0, I_m)$, the following holds:*

$$\lim_{m \to \infty} \sup_{t \in [0,T]} \|K(t) - K(0)\| > 0 \quad a.s.$$

*Proof.* The proof can be found in Appendix D. □

**Remark 3.6.** *It is worth noticing that our result holds under the assumption that a neural network with $m \to \infty$ can adequately approximate the solution $u^\star$ of the PDE* (1), *and that the training process is successful in achieving this approximation. The first assumption is justified by results such as the universal approximation theorem for neural networks [1]. Despite this optimistic training scenario, as demonstrated in Proposition 3.5, the constancy of the kernel is unattainable.*

|                                  | **Linear PDEs**         | **Nonlinear PDEs**                    |
|----------------------------------|-------------------------|---------------------------------------|
| NTK at initialization            | Deterministic           | Random *(Theorem 3.4)*                |
| NTK during training              | Constant                | Dynamic *(Proposition 3.5)*           |
| Hessian $H_r$                    | Sparse                  | Not sparse *(Proposition 3.7)*        |
| First-order convergence bound    | $\sim \lambda_{\min}(K)$ | $\sim 0$ or $\lambda_{\min}(K(t))$   |
| Second-order convergence bound   | $\sim 1$                | $\sim 0$ or $1$ *(Theorem 4.2)*       |

Table 1: Comparison of the theoretical results for linear and nonlinear PDEs.

In the context of nonlinear PDEs, converging to a linear regime is unattainable, even in the infinite-width limit, and this inability stems from the spectral norm of $H_r$, which is the Hessian of the residuals $r_\theta$ with respect to the parameters $\theta$. Indeed, in the linear scenario, the convergence of $\|H_r\|$ to 0 as $m \to \infty$ is crucial for demonstrating convergence to the linear regime, as established in Proposition B.3. Similar conclusions have been drawn in [22] for various deep learning architectures. However, we now show that this property does not hold for nonlinear PDEs.

**Proposition 3.7.** *Under Assumptions B.2 and 3.3 on the network and on the PDE, let us further assume that $R$ is a second-order polynomial. Then, the Hessian of the residuals $H_r$ is not sparse and*

$$\lim_{m \to \infty} \|H_r\| \geq \tilde{c},$$

*where the constant $\tilde{c}$ does not depend on $m$.*

*Proof.* The proof can be found in Appendix E, together with an explicit formula for $\tilde{c}$. □

**Remark 3.8.** *For the latter result, we additionally require that $R$ is a second-order polynomial, which includes many classic nonlinear PDEs like Burgers' or Navier-Stokes equations.*

We summarize all our results and provide a comparison with the linear case in Table 1. Motivated by the fact that the Hessian is not negligible, we shift our attention to second-order optimization methods and explore their convergence capabilities.

## 4 Convergence results

Before delving into second-order methods, let us revisit a convergence result for first-order ones. Traditional analyses of the gradient descent (6) often rely on the smoothness and convexity of the loss, assumptions that may not hold in the context of deep learning. As an alternative, numerous results concentrate on the infinite-width limit, particularly in connection with the NTK analysis. While we refrain from presenting a formal proof, we highlight the notable result below.

**Theorem 4.1.** *Under Assumption B.1 on the PDE and Assumption B.2 on the network defined by* (4), *consider the scenario where $m$ is sufficiently large. With high probability on the random initialization, there exists a constant $\mu > 0$, depending on the eigenvalues of $K$, such that gradient descent, employing a sufficiently small step size $\eta$, converges to a global minimizer of* (5) *with an exponential convergence rate, i.e.*

$$L(\theta(t)) \leq (1 - \eta\mu)^t L(\theta(0)).$$

*Proof.* See [8], [4], [22], and others. □

It is noteworthy that this result is presented at the level of gradient descent, i.e. the discretization of the gradient flow (6), which explains the constant $\eta$ representing its step size. Theorem 4.1 has also been extended to various types of architectures in [5]. We emphasize that this convergence result is rooted in the applicability of the Polyak-Lojasiewicz condition which, in turn, is linked to the smallest eigenvalue of the tangent kernel (denoted with $\lambda_{\min}$). In the case of linear PDEs, the tangent kernel $K(t)$ is positive definite [8] for any $t \in [0, T]$, leading to positive eigenvalues. The key finding in this context is that if $m$ is sufficiently large, $K(t) \approx \bar{K}$, where $\bar{K}$ is a deterministic matrix, which only

depends on the training input and not on the network's parameters $\theta$. As a result, in the infinite-width regime, the dynamics (8) can be approximated by

$$\begin{bmatrix} \partial_t u_\theta(\mathbf{x}^b) \\ \partial_t r_\theta(\mathbf{x}^r) \end{bmatrix} \approx -\bar{K} \begin{bmatrix} u_\theta(\mathbf{x}^b) - g(\mathbf{x}^b) \\ r_\theta(\mathbf{x}^r) \end{bmatrix}. \tag{11}$$

In the linear case, the key steps (i.e. the fact that the NTK is deterministic and constant) of the convergence proof of Theorem 4.1 cannot be adapted to nonlinear PDEs. Indeed, the stochasticity of the matrix and its dynamic behavior during training make the reasoning of [8] or [4] inapplicable, and it is challenging to show that the eigenvalues of $K(t)$ in the nonlinear case are uniformly bounded away from zero over training time. Nevertheless, we believe this question warrants further investigation.

Another issue linked to the NTK's eigenvalues is the phenomenon recognized as *spectral bias* by [39, 2, 29]. This is related to the fast decay of the NTK's eigenvalues, which characterize the rate at which the training error diminishes. The presence of small or unbalanced eigenvalues leads to slow convergence, particularly for high-frequency components of the PDE solution, or even to training failure. This occurs regardless of the linearity of the PDE differential operator $R$. In the next section, we show that under certain assumptions, second-order methods can help mitigate both problems.

## 4.1 Second-Order Optimization Methods

Due to all the aforementioned reasons and Proposition 3.7, our focus turns to the investigation of second-order optimization methods. These are powerful algorithms that leverage both the gradient and the Hessian of the loss function. Within this category, Quasi-Newton methods stand out as the most natural and widely known, relying on the Newton update rule

$$\theta(t + 1) = \theta(t) - \left[ \nabla^2 L(\theta(t)) \right]^{-1} \nabla L(\theta(t)). \tag{12}$$

However, the application of this update step relies on second-order derivatives, which are prohibitively expensive to compute as the number of parameters in the model increases. Indeed, the core idea behind Quasi-Newton methods involves utilizing an approximation of the Hessian as follows

$$\nabla^2 L(\theta) = J^T(t)J(t) + H_r r_{\theta(t)} \approx J^T(t)J(t) \tag{13}$$

in the formula (12). Here, $J(t) \in \mathbb{R}^{n \times p}$ represents the Jacobian of the loss at the training time $t$, and it aligns with the definition in (9). Since the Jacobian $J(t)$ is part of the evaluation of the gradient, the approximation (13) does not necessitate the computation of higher-order derivatives.

We now tackle the issues of spectral bias and slow convergence by presenting a result applicable to the Gauss-Newton method. In practice, when the number of parameters $p$ is larger than the number of samples $n$, the matrix $J^T(t)J(t)$ is surely singular. In this case, we consider the generalized inverse $(J^T(t)J(t))^\dagger$, instead of the inverse.

**Theorem 4.2.** *Consider the parameter $\theta(t)$ obtained by the Gauss-Newton flow below*

$$\partial_t \theta(t) = -(J^T(t)J(t))^\dagger \nabla L(\theta(t)). \tag{14}$$

*Then, the following holds*

$$\begin{bmatrix} \partial_t u_{\theta(t)}(\mathbf{x}^b) \\ \partial_t r_{\theta(t)}(\mathbf{x}^r) \end{bmatrix} = -U(t)D(t)U(t)^T \begin{bmatrix} u_{\theta(t)}(\mathbf{x}^b) \\ r_{\theta(t)}(\mathbf{x}^r) \end{bmatrix}, \tag{15}$$

*where $U(t) \in \mathbb{R}^{n \times n}$ is a unitary matrix and $D \in \mathbb{R}^{n \times n}$ is a diagonal matrix with entries $0$ or $1$. In particular, if $J(t)$ is full-rank for any $t \in [0, T]$, then convergence to a global minimum is attained.*

*Proof.* The proof is presented in Appendix F. $\square$

This result is significant as it indicates that when utilizing second-order methods via (14), convergence no longer depends on the eigenvalues of $K(t)$ as in (11), but rather on the elements of the diagonal matrix $D(t)$. Consequently, the training process becomes nearly spectrally unbiased, as the nonzero eigenvalues of the controlling matrix in (15) are all 1s. Let us now compare the cases of linear and nonlinear PDEs, in relation to the assumption of full-rankness of $J(t)$ and, consequently, the NTK.

- **Linear PDEs:** recent research [8] has theoretically confirmed that the NTK has full-rank in this case. Hence, convergence of second-order methods is achieved with *all* eigenvalues equal to 1, offering a notable advantage over (11) since the training method is unaffected by the spectral bias.

- **Nonlinear PDEs:** showing theoretically the full-rankness is a complicated task, particularly in light of Theorem 3.5, which highlights the stochastic and dynamic nature of the NTK. Similarly, verifying numerically the full-rankness of $J(t)$ is impractical due to the matrix's ill-conditioning, as mentioned in [40]. However, even if $J(t)$ is not full-rank, it holds that, although some singular values are zero, fast convergence for the remaining ones is attained.

Moreover, let us stress that the result in Theorem 4.2 applies to any network, including those with finite width. Thus, while the NTK model motivates the use of second-order methods, the key insights about spectral bias and convergence hold without assuming infinite width.

**Remark 4.3.** *In practice, the Gauss-Newton method becomes less computationally expensive when combined with inexact techniques such as Krylov subspace methods, conjugate gradient, BFGS, or LBFGS [27]. It has been shown that BFGS and LBFGS asymptotically approach the exact Hessian under certain conditions [21]. To extend our findings to more practical inexact methods, we can leverage these asymptotic convergence properties. However, while this approach is theoretically sound, the speed of convergence of quasi-Newton methods to the exact Newton method — specifically their matrix approximation accuracy — depends on the minimum eigenvalue of the Hessian [21][Theorem 6]. As discussed in our paper, the Hessian in PINNs is typically very poorly conditioned. As a result, quasi-Newton methods may require an impractically large number of training steps to converge to the true inverse Hessian and, thus, to begin training higher modes.*

# 5 Numerical Experiments

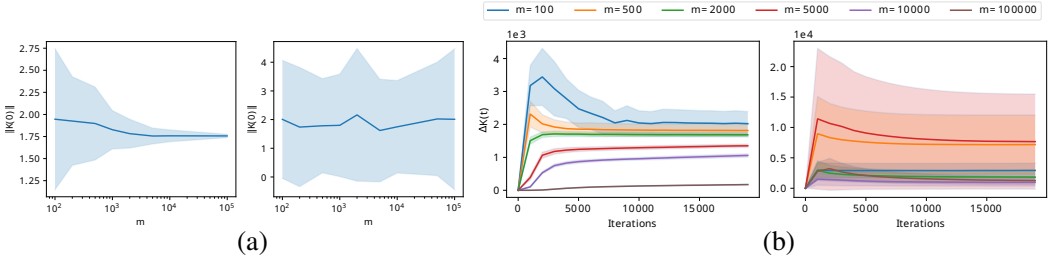

Figure 1: **(a)** Mean and standard deviation of the spectral norm of $K(0)$ as a function of the number of neurons $m$ for 10 independent experiments. **Left:** linear case. **Right:** nonlinear case. **(b)** Mean and standard deviation of $\Delta K(t) := \frac{\|K(t)-K(0)\|}{\|K(0)\|}$ over the network's width $m$, for 10 independent experiments. **Left:** linear case. **Right:** nonlinear case.

## 5.1 Empirical validation of our NTK results

First of all, we aim at numerically validate the results presented above, by comparing the NTK in case of linear and nonlinear PDEs. Our experiments are conducted on the following linear equation: $\partial_x^2 u(x) = \frac{16}{\pi^2} \sin(\frac{4}{\pi}x)$. Meanwhile, as nonlinear PDE, we consider $u(x)\partial_x u(x) = \frac{16}{\pi^2} \sin(\frac{4}{\pi}x)$. Notably, these results exhibit consistency across various equations and experimental setups.

The result in Theorem 3.4 is confirmed by the numerical experiments depicted in Figure 1, part (a): in the linear case the NTK at initialization converges to a deterministic matrix when $m \to \infty$, while this does not happen in the nonlinear case. The statement of Proposition 3.5 is confirmed in part (b) of Figure 1 by showing that the constancy of the NTK during training is not attainable in the nonlinear case. Moreover, the result in Proposition 3.7 is supported by part (a) of Figure 2, where we compare the sparsity of the Hessian at initialization $H_r(0)$ in both the linear and nonlinear case. Moreover, we observe that in the linear scenario $\|H_r\|$ decays as $m$ grows, contrarily to the nonlinear example. Similarly, we refer to Figure 2, part (b) for a comparison of the eigenvalues when training with first-order or second-order methods on Burgers' equation.

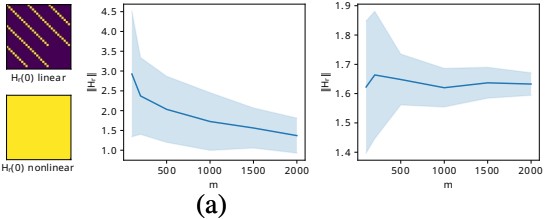
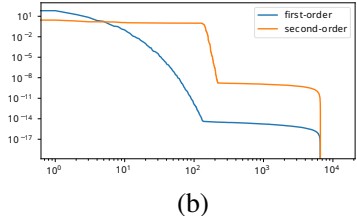

|       (a)       |       (b)       |

Figure 2: **(a) Left:** in yellow the non-zero components of the Hessian matrix at initialization (up in the linear case, down the nonlinear one). **Center:** mean and standard deviation of the spectral norm of the $H_r(0)$ over $m$ in the linear case (for 10 independent experiments). **Right:** same as Center, but for a nonlinear PDE. **(b)** Eigenvalues of $K(0)$ for a first-order optimizer and $D(0)$ for a second-order method applied to Burgers' equation.

## 5.2 Employment of second-order methods

Among all second-order methods, in our numerical experiments we make use of an existing variant of the Levenberg-Marquardt (LM) algorithm, as it offers further stability through the update rule

$$\theta(t+1) = \theta(t) - \left[J^T(t)J(t) + \lambda \mathrm{Id}_p\right]^{-1} \nabla L(\theta(t)),$$

where $\lambda$ is a damping parameter adjusted by the algorithm. In practice, the iterative step of LM can be considered as an average, weighted by $\lambda$, between the Gradient Descent step and a Gauss-Newton method. This aspect of the LM algorithm represents its crucial advantage over other Quasi-Newton methods such as Gauss-Newton or BFGS. Indeed, Quasi-Newton methods show good performance when the initial guess of the solution $u_\theta$ is close to the correct one. The update rule of LM avoids this issue by relying on simil-gradient descent steps at early iteration. Moreover, the parameter $\lambda$ typically decreases during training, in order to converge to a Quasi-Newton method when close to the optimum. Our primary aim is to showcase the effectiveness of second-order methods for nonlinear PINNs, a point which has been supported by findings such as those in [26]: their approach also employs a second-order method, akin to a Gauss-Newton method in function spaces. For details on the modified LM algorithm, along with pseudocode, we refer to Appendix G.

**Details on the Networks**  The neural network architectures adopted in the experiments are standard Vanilla PINNs with hyperbolic tangent as activation function. All of the PINNs trained in our analysis are characterized by 5 hidden layers with 20 neurons each. Every training is performed for 10 independent neural networks initialized with Xavier normal distribution [10]. All models are implemented in PyTorch [28] and trained on a single NVIDIA A10 GPU.

**Test Cases**  We assess our theoretical findings on the following equations:

- *Wave/Poisson/Convection Equation*: despite being linear PDEs, they represent a suitable scenario to showcase the detrimental effect of the spectral bias on the training of PINNs, due to the presence of high-frequency components in the solution.

- *Burgers' Equation*: this nonlinear PDE is commonly used to test PINNs, and usually they reach a valid solution even with a first-order optimizer, due to the PDE's simplicity.

- *Navier-Stokes Equation*: it poses challenges for both PINNs and classical methods, being a difficult nonlinear PDEs. We test the case of the fluid flow in the wake of a 2D cylinder [17].

For the sake of compactness, we refer to Appendix G for detailed descriptions of the mentioned PDEs, and to Appendix H for supplementary numerical experiments not included in the main text. We compare results obtained by the LM algorithm with those from commonly used optimizers for training PINNs, such as Adam [18] and L-BFGS [24]. Where not stated otherwise, Adam is trained for $10^5$ iterations and LM for $10^3$ iterations. Additionally, we provide a comparison with other methods that are ad-hoc enhancements of PINNs, such as loss balancing [40] (also known as NTK rescaling), Random Fourier Features (RFF) [39], and curriculum training (CT) [19]. Our performance metric is the relative $L^2$ loss on the test set, detailed in Appendix H formula (28).

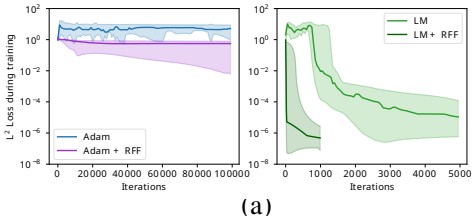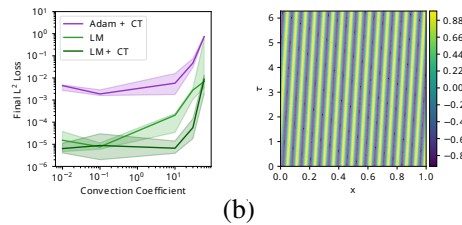

(a) (b)

Figure 3: **(a)** Poisson equation: median and standard deviation of the relative $L^2$ loss for different optimizers over training iterations (repetitions over 10 independent runs). **(b)** Convection equation: median and standard deviation of the $L^2$ loss after 1000 iterations achieved over 5 independent runs with and without CT for different values of the convection coefficient $\beta$ **(left)** and solution obtained with LM (and no other enhancement) after 5000 iterations with $\beta = 100$ **(right)**.

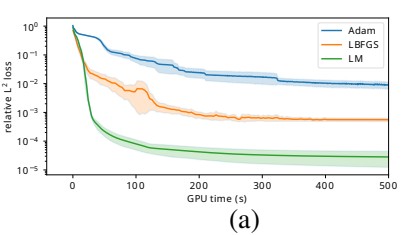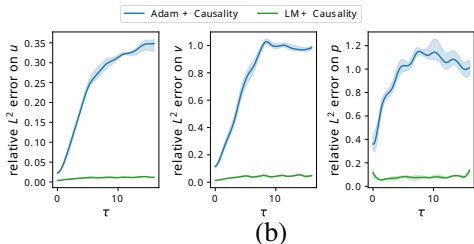

(a) (b)

Figure 4: **(a)** Burgers' equation: mean and standard deviation of the relative $L^2$ loss for various optimizers over wall time (repetitions over 10 independent runs). **(b)** Navier-Stokes equation: mean and standard deviation of the relative $L^2$ loss over the PDE time $\tau$ for PINNs trained with Adam and LM (10 independent runs). Both optimization methods are enhanced with causality training.

**Linear PDEs affected by spectral bias**   In Figure 3, we demonstrate the effectiveness of second-order methods in handling equations with high spectral bias. Part (a) of Figure 3 focuses on the Poisson equation with high-frequency components, for which is common to use RFF [39]. On the left, we show that Adam requires RFF to converge to a reasonable solution. On the right, we observe that LM not only significantly outperforms Adam combined with RFF, but also that incorporating RFF with LM leads to remarkable loss reduction from the very first iterations. In Part (b) of Figure 3, we investigate the effect of high convection coefficients $\beta$ in the convection equation as discussed in [19], where it is shown that a PINN trained with Adam necessitates of curriculum training to achieve meaningful results on such a spectrally biased PDE. However, we show on the left Figure 3, part (b), that the LM optimizer can handle higher values of $\beta$, especially when curriculum training is introduced. Remarkably, on the right of Figure 3(b), we show that a PINN trained with LM, without any other enhancements, achieves high accuracy with $\beta$ values up to 100. This level of accuracy is not feasible with Adam and curriculum training alone, which, as noted in [19], manages coefficients only up to 20.

**Nonlinear PDEs**   Firstly, we consider the case of Burgers' equation, where convergence is achievable even with first-order methods. To address concerns about the additional computational time required by second-order methods, in Figure 4, part (a), we display the relative $L^2$ loss over wall time when training on Burgers' equation. All training methods can reach a reasonable solution, however, while the precision of PINNs trained with Adam and L-BFGS is approximately $10^{-3}$, PINNs trained with LM can consistently attain precision around $10^{-5}$ in few iterations and very short GPU time. Figure 4 also provides a qualitative estimate of the runtime of LM in comparison to Adam and L-BFGS. The intermediate performance of L-BFGS, falling between first- and second-order methods, is explained in Remark 4.3. Lastly, a similar outcome can be seen in part (b) of Figure 4, where we demonstrate that employing the LM optimizer makes it possible to obtain a reasonable solution even for Navier-Stokes equation in terms of relative $L^2$ loss over PDE time. Notice that in this case, we employ causality training [41] for both Adam and LM.

### 5.3 Limitations and possible solutions

The major limitation of our findings is related to scalability. Traditionally, second-order methods have been avoided for machine learning models due to their poor scaling with an increasing number of parameters. However, one can adopt classical PDE solution approaches, such as domain decomposition, to utilize a collection of smaller networks instead of a single large one. Similarly, one can embrace machine learning-based solutions such as ensemble models [12] or mixture of experts [11]. We advocate that existing models such as [14, 25, 37] could already be strongly enhanced with the usage of second-order methods for training. In the scenario where these approaches are impractical, one could also resort to techniques in the field of optimization to enable the scalability of the method. For medium to large-sized networks, the challenge of storing the matrix $J^T J$ in GPU memory becomes infeasible. This can be addressed through an inexact LM method, which involves solving the equivalent system $\|J\theta - r_\theta\| = 0$ using a Krylov subspace iterative method (LSQR or LSMR) [27, 7]. These methods only require Jacobian-vector products, which can be efficiently computed through backpropagation.

## 6 Conclusion

In this paper, we conduct an in-depth analysis of PINNs training utilizing the NTK framework. We elucidate the distinction between linear and nonlinear cases, and reveal that even in the optimistic infinite-width limit, favorable outcomes observed with NTK in linear cases do not extend to nonlinear PDEs. Motivated by the NTK anaylsis, we emphasize the significant advantage of employing second-order methods. These seem to mitigate the spectral bias issue and to improve convergence even for challenging nonlinear PDEs. Second-order methods, such as LM, consistently achieve a precision comparable or even better than the state-of-the-art presented in [13]. Notably, our findings demonstrate that convergence is attainable without resorting to typical training protocols aimed at enhancing PINNs. However, combining these enhancements with second-order training methods can further improve accuracy while reducing computational time, as demonstrated in our numerical experiments. Accuracy and convergence guarantees are indeed two crucial components for the majority of real-world applications of PDE solvers. In practice, second-order methods may be preferable when the solution contains high frequencies, when the application demands high accuracy, or when the target PDE is nonlinear. A key objective of our paper is to highlight that, despite their scalability challenges, second-order methods could help bridge the gap between black-box machine learning models and PDE solutions in scientific machine learning.

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

## Supplemental Material

This supplemental material is divided into the following eight appendices.

- Appendix A: Details about the NTK Matrix

- Appendix B: Standard NTK results for linear PDEs

- Appendix C: Proof of Theorem 3.4

- Appendix D: Proof of Proposition 3.5

- Appendix E: Proof of Proposition 3.7

- Appendix F: Proof of Theorem 4.2

- Appendix G: Details about the Numerical Experiments

- Appendix H: Further Numerical Experiments

In the following we denote with $\| \cdot \|_2$ and $\langle \cdot, \cdot \rangle$ the Euclidean and scalar product on $\mathbb{R}^d$, respectively. The Euclidean ball centered in $x$ with radius $R$ is indicated with $B(x, R)$. We denote with $\| \cdot \|$ the spectral norm of a matrix and with $\mathrm{I}_n$ the identity matrix of dimension $n \times n$.
We abbreviate with i.i.d. independently and identically distributed random variables. $\mathbb{E}[X]$ denotes the mean of the random variable $X \in \mathbb{R}^d$, while $\mathrm{Cov}[\mathrm{X}]$ is its covariance matrix. Convergence of $X_n$ to $X$ in distribution is indicated with $X_n \xrightarrow{\mathcal{D}} X$, while convergence in probability with $X_n \xrightarrow{\mathcal{P}} X$. $\mathcal{GP}$ denotes a Gaussian Process.
The operator $\nabla$ denotes the gradient of a function on $\mathbb{R}^d$, while $\partial_x f(x, y)$ the partial derivative of $f$ with respect to the variable $x$.

## A    Details about the NTK Matrix

We define the following matrices

$$
\partial_\theta u_\theta(x) = [\partial_{\theta_1} u_\theta(x) \quad \cdots \quad \partial_{\theta_m} u_\theta(x)],
$$
$$
\partial_\theta r_\theta(x) = [\partial_{\theta_1} r_\theta(x) \quad \cdots \quad \partial_{\theta_m} r_\theta(x)],
$$
$$
\partial_\theta \Phi[u_\theta](x) = \begin{bmatrix} \partial_{\theta_1} \Phi_1[u_\theta](x) & \cdots & \partial_{\theta_m} \Phi_1[u_\theta](x) \\ \vdots & \cdots & \vdots \\ \partial_{\theta_1} \Phi_k[u_\theta](x) & \cdots & \partial_{\theta_m} \Phi_k[u_\theta](x) \end{bmatrix}.
$$

By $\partial_\theta u_\theta(\mathbf{x}^b)$, $\partial_\theta r_\theta(\mathbf{x}^r)$ and $\partial_\theta \Phi[u_\theta](\mathbf{x}^r)$ we mean the same matrices as before, calculated in each $x_i^b$ ($x_i^r$ respectively) and stacked vertically, e.g.:

$$
\partial_\theta \Phi[u_\theta](\mathbf{x}^b) = \begin{bmatrix} \partial_\theta \Phi[u_\theta](x_1^b) \\ \vdots \\ \partial_\theta \Phi[u_\theta](x_{N_b}^b) \end{bmatrix}.
$$

The only exception is given by:

$$
\nabla R(\Phi[u_\theta](\mathbf{x}^r)) = \begin{bmatrix} \nabla R(\Phi[u_\theta](x_1^r)) & \mathbf{0} & \cdots & \cdots & \mathbf{0} \\ \mathbf{0} & \nabla R(\Phi[u_\theta](x_2^r)) & \mathbf{0} & \cdots & \mathbf{0} \\ \vdots & \cdots & \cdots & \cdots & \vdots \\ \mathbf{0} & \cdots & \cdots & \mathbf{0} & \nabla R(\Phi[u_\theta](x_{N_r}^r)) \end{bmatrix}. \tag{16}
$$

While the Hessians have the following structure:

$$H_u(x) := \begin{bmatrix} \partial^2_{\theta_1 \theta_1} u_\theta(x) & \cdots & \partial^2_{\theta_1 \theta_m} u_\theta(x) \\ \vdots & \ldots & \vdots \\ \partial^2_{\theta_m \theta_1} u_\theta(x) & \cdots & \partial^2_{\theta_m \theta_m} u_\theta(x) \end{bmatrix},$$

$$H_r(x) := \begin{bmatrix} \partial^2_{\theta_1 \theta_1} r_\theta(x) & \cdots & \partial^2_{\theta_1 \theta_m} r_\theta(x) \\ \vdots & \ldots & \vdots \\ \partial^2_{\theta_m \theta_1} r_\theta(x) & \cdots & \partial^2_{\theta_m \theta_m} r_\theta(x) \end{bmatrix}, \tag{17}$$

$$H_{\Phi_i}(x) := \begin{bmatrix} \partial^2_{\theta_1 \theta_1} \Phi_i[u_\theta](x) & \cdots & \partial^2_{\theta_1 \theta_m} \Phi_i[u_\theta](x) \\ \vdots & \ldots & \vdots \\ \partial^2_{\theta_m \theta_1} \Phi_i[u_\theta](x) & \cdots & \partial^2_{\theta_m \theta_m} \Phi_i[u_\theta](x) \end{bmatrix}.$$

# B  NTK for linear PDEs

First of all, we list here all the assumptions needed on the differential operator $\mathcal{R}$ and the neural network in (4).

**Assumption B.1** (on $\mathcal{R}$). *The differential operator $\mathcal{R}$ is linear, which implies that $R$ is linear.*

**Assumption B.2** (on the network). *Given the network* (4)*, we assume the following properties:*

(i) *there exists a constant $C > 0$ such that all parameters of the network are uniformly bounded for $t \in [0, T]$,*

$$\sup_{t \in [0,T]} ||\theta(t)||_\infty \leq C \quad \text{with } C \text{ independent from } m.$$

(ii) *there exists a constant $C > 0$ such that*

$$\int_0^T \left| \sum_{i=1}^{N_b} (u_{\theta(\tau)}(x_i^b) - g(x_i^b)) \right| d\tau \leq C,$$

$$\int_0^T \left| \sum_{i=1}^{N_r} (\Phi[u_{\theta(\tau)}](x_i^r)) \right| d\tau \leq C.$$

(iii) *the activation function $\sigma$ and as well as its derivatives $\sigma^{(i)}$ up to power $k + 1$ are smooth and $|\sigma^{(i)}| \leq C$ for $i = 1, \ldots, k$, where $\sigma^{(i)}$ denotes the $i$-th order derivative of $\sigma$.*

In order to present the results, we denote with $H_r$ the Hessian of the residuals $r_\theta$ with respect to the parameters $\theta$. The Hessian plays an important role in Proposition B.3, which aims to list all the prior results that can be derived by combining Theorem 4.4 of [40], Theorem 3.2 of [22].

**Proposition B.3.** *Consider a fully-connected neural network given by* (4)*, under the Assumption B.2 on the network and Assumption B.1 on the PDE. For the minimization of the loss function* (5) *through gradient flow, starting from a Gaussian random initialization $\theta(0)$, it holds that for any $T > 0$,*

- *the randomly initialized tangent kernel $K(0)$ converges in probability to a deterministic kernel $\tilde{K}$ as $m \to \infty$;*

- *the Hessian matrix $H_r$ of the residuals is sparse and*

$$||H_r|| = O\left( \frac{1}{\sqrt{m}} \right),$$

*hence the spectral norm converges to 0 as $m \to \infty$;*

- *as a consequence, the NTK is nearly constant during training, i.e.*

$$\lim_{m \to \infty} \sup_{t \in [0,T]} ||K(t) - K(0)||_2 = 0;$$

*Proof.* The proof can be found in the papers mentioned above or as a special (linear) case in Appendix C-E. □

# C Proof of Proposition 3.4.

First of all, we derive a result about the behavior of the vector of partial derivatives $\Phi[u]$. The Proposition C.1 below is a generalization of Theorem 4.1 in [40] for any derivative of order $k$. This means that there are no nonlinearities involved, since these are encoded in the function $R$. Moreover we study the full vector and not each component separately as it is done in [40]. This is needed in the following proofs.

**Proposition C.1.** *Consider a fully-connected neural network of one hidden layer as in* (4)*, under Assumption B.2. Then, starting from $\theta(0)$ i.i.d. from $\mathcal{N}(0, \mathrm{Id})$, it holds that*

$$\Phi[u_{\theta(0)}](x) \xrightarrow{\mathcal{D}} \mathcal{GP}(0, \Sigma(x, x')) \quad \text{for any } x, x' \in \Omega,$$

*as $m \to \infty$, where $\mathcal{D}$ means convergence in distribution and $\Sigma$ is explicitly calculated.*

*Proof.* To ease the notation, we omit the initial time $0$ and denote $u_{\theta(0)}$ with $u_\theta$. Similarly, all the weights matrices and biases $W^1(0), W^0(0), b^1(0), b^0(0)$ are indicated with $W^1, W^0, b^1, b^0$. Now according to the definition of $\Phi$ and the fact that it is linear, we obtain that

$$\Phi[u_\theta](x) = \frac{1}{\sqrt{m}} W^1 \cdot \Phi[\sigma(W^0 x + b^0)] = \frac{1}{\sqrt{m}} \sum_{j=1}^m W_j^1 \Phi[\sigma(W_j^0 x + b_j^0)].$$

According to our assumptions, $W_j^1 \Phi[\sigma(W_j^0 x + b_j^0)]$ are i.i.d. random variables. We prove below that their moments are finite, hence by the multidimensional Central Limit theorem (CLT) we can conclude that, for every $x \in \Omega$,

$$\Phi[u_\theta](x) \xrightarrow{\mathcal{D}} \mathcal{N}(0, \Gamma(x)),$$

with covariance matrix:

$$\Gamma(x) = \mathrm{Cov}_{u,v \sim \mathcal{N}(0,1)} \left[ \Phi[\sigma(ux + v)] \right].$$

Now we compute the covariance of the limit gaussian process. In order to do so, we first need to show that $\Phi_i[u_\theta](x)$ are uniformly integrable with respect to $m$ for every $i = 1, ..., k$. It follows from:

$$\sup_m \mathbb{E}[|\Phi_i[u_\theta](x)|^2] = \sup_m \mathbb{E} \left[ \frac{1}{m} \sum_{j,l=1}^m W_j^1 W_l^1 \Phi_i[\sigma(W_j^0 x + b_j^0)] \Phi_i[\sigma(W_l^0 x' + b_l^0)] \right]$$

$$= \sup_m \mathbb{E} \left[ \frac{1}{m} \sum_{j=0}^m (W_j^1)^2 \Phi_i[\sigma(W_k^0 x + b_k^0)]^2 \right] = \mathbb{E} \left[ \Phi_i[\sigma(W_j^0 x + b_j^0)]^2 \right]$$

$$\leq C^2 \tau^2,$$

where $C = \max_{1 \leq i \leq k} ||\sigma^{(i)}||_\infty$ and $\sigma^{(i)}$ indicates the $i$-th order derivative of $\sigma$, while $\tau = \max_{1 \leq i \leq k} \mathbb{E}_{y \sim \mathcal{N}(0,1)}[|y|^i] < \infty$.

Now, for any given point $x, x' \in \Omega$ we have that

$$\Sigma(x, x')_{i,j} = \lim_{m \to \infty} \mathbb{E} \left[ \Phi_i[u_\theta](x) \Phi_j[u_\theta](x') \right] =$$

$$= \lim_{m \to \infty} \mathbb{E} \left[ \frac{1}{m} \sum_{l_1, l_2 = 1}^m W_{l_1}^1 W_{l_2}^1 \Phi_i[\sigma(W_{l_1}^0 x + b_{l_1}^0)] \Phi_j[\sigma(W_{l_2}^0 x' + b_{l_2}^0)] \right] =$$

$$= \lim_{m \to \infty} \mathbb{E} \left[ \frac{1}{m} \sum_{l=1}^m (W_l^1)^2 \Phi_i[\sigma(W_l^0 x + b_l^0)] \Phi_j[\sigma(W_l^0 x' + b_l^0)] \right] =$$

$$= \mathbb{E}_{u,v \sim \mathcal{N}(0,1)} \left[ \Phi_i[\sigma(ux + v)] \Phi_j[\sigma(ux' + v)] \right].$$

$\square$

**Lemma C.2.** *Consider a fully-connected neural network of one hidden layer as in* (4)*, under Assumption B.2. Let us define*

$$K_\Phi(0) = \begin{bmatrix} \partial_\theta u_{\theta(0)}(\mathbf{x}^b) \\ \partial_\theta \Phi[u_{\theta(0)}](\mathbf{x}^r) \end{bmatrix} \begin{bmatrix} \partial_\theta u_{\theta(0)}(\mathbf{x}^b)^T & \partial_\theta \Phi[u_{\theta(0)}](\mathbf{x}^r)^T \end{bmatrix},$$

*where $\Phi$ is the collection of all the partial derivatives of $u$, as in* (3)*, and $\theta(0) \sim \mathcal{N}(0, \mathrm{Id})$ i.i.d.. It follows that $K_\Phi(0)$ converges in probability to a deterministic limiting kernel as $m \to \infty$.*

*Proof.* The component $\partial_\theta u_{\theta(0)}$ is linear, hence it is standard as in [40], Lemma 3.1. While the rest of the matrix needs to be generalized to any derivative $\Phi_i$ for $i = 1, \ldots, k$.
For any $i, j = 1, \ldots, k$ and every $x, x' \in \Omega$ consider each entry

$$\partial_\theta \Phi_i[u_{\theta(0)}](x) \ \partial_\theta \Phi_j[u_{\theta(0)}](x')^T = \sum_{l=1}^{4m} \partial_{\theta_l} \Phi_i[u_{\theta(0)}](x) \ \partial_{\theta_l} \Phi_j[u_{\theta(0)}](x')$$

$$= \sum_{l=1}^{4m} \Phi_i \left[ \partial_{\theta_l} u_{\theta(0)} \right](x) \ \Phi_j \left[ \partial_{\theta_l} u_{\theta(0)} \right](x')$$

where the second equality follows from Schwarz theorem (because of the smoothness of the derivatives of $u$), and the linearity of the operator $\Phi$. This sum has to be split in 4 parts, one for each possible type of $\theta_l$ (in $W^1$, $W^0$, $b^0$ or $b^1$). Here we present the case when $\theta_l = W_l^1$, while the other cases are analogous:

$$\sum_{l=1}^{m} \Phi_i \left[ \partial_{W_l^1} u_{\theta(0)} \right](x) \ \Phi_j \left[ \partial_{W_l^1} u_{\theta(0)} \right](x') =$$

$$= \frac{1}{m} \sum_{l=1}^{m} \Phi_i \left[ \sigma(W_l^0(0)x + b_l^0(0)) \right] \ \Phi_j \left[ \sigma(W_l^0(0)x' + b_l^0(0)) \right]$$

$$\xrightarrow{\mathcal{P}} \mathbb{E}_{u,v \sim \mathcal{N}(0,1)} \left[ \Phi_i[\sigma(ux + v)] \ \Phi_j[\sigma(ux' + v)] \right],$$

and the limit in probability in the last line comes from the law of Large Numbers. $\square$

**Lemma C.3.** *Suppose that there exist $R > 0$ and $\epsilon > 0$ such that $\forall \theta \in B(\theta(0), R)$ it holds*

$$\|H_u(\mathbf{x}^b)\| < \epsilon,$$

$$\|H_{\Phi_j}(\mathbf{x}^b)\| < \epsilon \quad \forall j = 1, ..., k.$$

*Then $\max_{\theta \in B(\theta_0, R)} \|K_\Phi(t) - K_\Phi(0)\| = O(\epsilon R)$.*

*Proof.* Using the properties of the spectral norm, we just need to bound each block of $J(t)$ as follows

$$\|J(t) - J(0)\| \leq \sum_{i=0}^{N_r} \|\partial_\theta \Phi[u_{\theta(t)}](x_i^r) - \partial_\theta \Phi[u_{\theta(0)}](x_i^r)\| + \sum_{i=0}^{N_b} \|\partial_\theta u_{\theta(t)}(x_i^b) - \partial_\theta u_{\theta(0)}(x_i^b)\|$$

$$\leq k N_r \max_{i,j} \|\partial_\theta \Phi_j[u_{\theta(t)}](x_i^r) - \partial_\theta \Phi_j[u_{\theta(0)}](x_i^r)\|$$

$$+ N_b \max_i \|\partial_\theta u_{\theta(t)}(x_i^b) - \partial_\theta u_{\theta(0)}(x_i^b)\|$$

$$\leq k N_r \max_{i,j} \left( \max_{\theta \in B(\theta(0), R)} \|H_{\Phi_j}(x_i^r)\| \right) \|\theta - \theta_0\|$$

$$+ N_b \max_i \left( \max_{\theta \in B(\theta(0), R)} \|H_u(x_i^b)\| \right) \|\theta - \theta_0\|$$

$$\leq \max(k N_r, N_b) \epsilon R$$

Hence:

$$\|K_\Phi(t) - K_\Phi(0)\| = \|J(t)J(t)^T - J(0)J(0)^T\| \leq \|J(t) - J(0)\| \cdot (\|J(t)\| + \|J(0)\|)$$

$$\leq \max(k N_r, N_b) \epsilon R(\|J(t)\| + \|J(0)\|)$$

and the last norm is bounded on $B(\theta(0), R)$ by smoothness of the model. $\square$

**Lemma C.4.** *Under Assumption B.1 on the PDE and Assumption B.2 on the network, then $K_\Phi$ is nearly constant during training, i.e.*

$$\lim_{m\to\infty} \sup_{t\in[0,T]} \|K_\Phi(t) - K_\Phi(0)\| = 0.$$

*Proof.* The statement follows by combining Lemma C.3 and Lemma E.1. □

Now we are in position to prove Theorem 3.4:

*Proof.* (of Theorem 3.4) By using the chain rule on the residual term, we can explicitly compute:

$$K(0) = \begin{bmatrix} \partial_\theta u_\theta(\mathbf{x}^b) \\ \partial_\theta r_\theta(\mathbf{x}^r) \end{bmatrix} \begin{bmatrix} \partial_\theta u_\theta(\mathbf{x}^b)^T & \partial_\theta r_\theta(\mathbf{x}^r)^T \end{bmatrix} =$$

$$= \begin{bmatrix} \partial_\theta u_\theta(\mathbf{x}^b) \\ \nabla R(\Phi[u_\theta](\mathbf{x}^r))\partial_\theta\Phi[u_\theta](\mathbf{x}^r) \end{bmatrix} \begin{bmatrix} \partial_\theta u_\theta(\mathbf{x}^b)^T & \nabla R(\Phi[u_\theta](\mathbf{x}^r))\partial_\theta\Phi[u_\theta](\mathbf{x}^r)^T \end{bmatrix} =$$

$$= \underbrace{\begin{bmatrix} \mathrm{Id} & 0 \\ 0 & \nabla R(\Phi[u_\theta](\mathbf{x}^r)) \end{bmatrix}}_{\Lambda_R(0)} \underbrace{\begin{bmatrix} \partial_\theta u_\theta(\mathbf{x}^b) \\ \partial_\theta\Phi[u_\theta](\mathbf{x}^r) \end{bmatrix} \begin{bmatrix} \partial_\theta u_\theta(\mathbf{x}^b)^T & \partial_\theta\Phi[u_\theta](\mathbf{x}^r)^T \end{bmatrix}}_{K_\Phi(0)} \underbrace{\begin{bmatrix} \mathrm{Id} & 0 \\ 0 & \nabla R(\Phi[u_\theta](\mathbf{x}^r))^T \end{bmatrix}}_{\Lambda_R(0)^T},$$

where we have denoted $\theta(0)$ with $\theta$ and omitted the initial time step and $\nabla R(\Phi[u_\theta](\mathbf{x}^r))$ is defined in (16). Let us first observe that the linear part, i.e. $K_\Phi(0)$, converges in probability to a deterministic limit by Lemma C.2. Moreover, $\Phi[u_{\theta(0)}]$ converges in distribution to a gaussian process by Proposition C.1. Regarding the nonlinear part denoted with $\Lambda_R(0)$, we know by assumption that $\nabla R$ is a continuous function, hence we can apply the Continuous Mapping Theorem and conclude that

$$\nabla R(\Phi[u_\theta](x)) \xrightarrow{\mathcal{D}} \nabla R\left(\mathcal{GP}(0, \Sigma(x, x'))\right) \quad \text{for } x, x' \in \Omega.$$

From this, the convergence of $K(0)$ follows by Slutsky's theorem. □

# D  Proof of Proposition 3.5

*Proof.* Recall that we denote with $K(t)$ the NTK obtained with $\theta(t)$, evolving according to the gradient flow (6). Similarly, $K(0)$ is the NTK at initialization, i.e. with $\theta(0) \sim \mathcal{N}(0, \mathrm{Id}_m)$. We can rewrite the kernels in terms of their linear and nonlinear part as we did for the proof of Theorem 3.4, and obtain

$$\lim_{m\to\infty} \sup_{t\in[0,T]} \|K(t) - K(0)\| \geq \lim_{m\to\infty} \|K(t_m) - K(0)\|$$

$$= \lim_{m\to\infty} \|\Lambda_R(t_m)K_\Phi(t_m)\Lambda_R(t_m)^T - \Lambda_R(0)K_\Phi(0)\Lambda_R(0)^T\|$$

$$\geq \lim_{m\to\infty} \Big| \|\Lambda_R(t_m)K_\Phi(0)\Lambda_R(t_m)^T - \Lambda_R(0)K_\Phi(0)\Lambda_R(0)^T\|$$

$$- \|\Lambda_R(t_m)[K_\Phi(t) - K_\Phi(0)]\Lambda_R(t_m)^T\| \Big|,$$

where the last is obtained by applying the inverse triangular inequality, after summing and subtracting the needed terms. Moreover, by considering that $\sup_{t\in[0,T]} \|K_\Phi(t) - K_\Phi(0)\| \to 0$ as $m \to \infty$ by Lemma C.4, we obtain that

$$\lim_{m\to\infty} \sup_{t\in[0,T]} \|K(t) - K(0)\| \geq \lim_{m\to\infty} \|\Lambda_R(t_m)K_\Phi(0)\Lambda_R(t_m)^T - \Lambda_R(0)K_\Phi(0)\Lambda_R(0)^T\| =$$

$$= \lim_{m\to\infty} \left\| \begin{bmatrix} \mathrm{Id} & 0 \\ 0 & \nabla R(\Phi[u(t_m)]) \end{bmatrix} K_\Phi(0) \begin{bmatrix} \mathrm{Id} & 0 \\ 0 & \nabla R(\Phi[u(t_m)]) \end{bmatrix}^T \right.$$

$$\left. - \begin{bmatrix} \mathrm{Id} & 0 \\ 0 & \nabla R(\Phi[u_{\theta(0)}]) \end{bmatrix} K_\Phi(0) \begin{bmatrix} \mathrm{Id} & 0 \\ 0 & \nabla R(\Phi[u_{\theta(0)}]) \end{bmatrix}^T \right\|.$$

Observe that (10) implies that $\Phi[u(t_m)] \to \Phi[u^\star]$, hence $\nabla R(\Phi[u_{\theta(t)]}) \to \nabla R(\Phi[u^\star])$ as $m \to \infty$ by continuity of $\nabla R$. Combining this and Lemma C.2, we find

$$\lim_{m \to \infty} \left\| \begin{bmatrix} Id & 0 \\ 0 & \nabla R(\Phi[u(t_m)]) \end{bmatrix} K_\Phi(0) \begin{bmatrix} Id & 0 \\ 0 & \nabla R(\Phi[u(t_m)]) \end{bmatrix}^T \right.$$

$$\left. - \begin{bmatrix} Id & 0 \\ 0 & \nabla R(\Phi[u_{\theta(0)}]) \end{bmatrix} K_\Phi(0) \begin{bmatrix} Id & 0 \\ 0 & \nabla R(\Phi[u_{\theta(0)}]) \end{bmatrix}^T \right\|$$

$$= \left\| \begin{bmatrix} Id & 0 \\ 0 & \nabla R(\Phi[u^\star]) \end{bmatrix} K_\Phi(0) \begin{bmatrix} Id & 0 \\ 0 & \nabla R(\Phi[u^\star]) \end{bmatrix}^T \right.$$

$$\left. - \begin{bmatrix} Id & 0 \\ 0 & \nabla R(\Phi[u_{\theta(0)}]) \end{bmatrix} K_\Phi(0) \begin{bmatrix} Id & 0 \\ 0 & \nabla R(\Phi[u_{\theta(0)}]) \end{bmatrix}^T \right\|.$$

Finally, to prove our statement, we just need to show that the matrix above is not $0$ almost surely, or at least one of its components. Let us fix a collocation point $x \in \Omega$ and let us define the function $f : \mathbb{R}^k \to \mathbb{R}$:

$$f(w) := \nabla R(\Phi[u^\star](x)) K_\Phi(0)_{(x,x)} \nabla R(\Phi[u^\star](x))^T - \nabla R(w) K_\Phi(0)_{(x,x)} \nabla R(w)^T, \quad (18)$$

where $K_\Phi(0)_{(x,x)}$ denotes the kernel evaluation at a fixed collocation point. The first term on the right hand side of (18) is a deterministic vector, so $f$ is a well defined deterministic analytic function. Moreover, if $R$ is nonlinear, $f$ is not identically zero.

By the properties of analytic functions we can conclude that $Leb(\{w \in \mathbb{R}^k | f(w) = 0\}) = 0$, where $Leb$ denotes the Lebesgue measure. Notice that $\Phi[u_{\theta(0)}](x) \sim \mathcal{N}(0, \Sigma(x))$ in the infinite-width limit as proven in Proposition C.1 and a consequence of that proof is that $\Sigma(x)$ is not singular. This implies that

$$\mathbb{P}(f(\Phi[u_{\theta(0)}](x)) = 0) = 0.$$

$\square$

# E Proof of Proposition 3.7

We present here some preparatory results.

**Lemma E.1.** *For any $i = 1 \ldots k$ and any $x \in \Omega$, the Hessian $H_{\Phi_i}(x)$ as defined in (17) is such that*

$$\|H_{\Phi_i}(x)\| = O(\frac{1}{\sqrt{m}}).$$

*Proof.* Recall that

$$(H_{\Phi_i}(x))_{jl} = \partial^2_{\theta_j \theta_l} \Phi_i[u_\theta](x), \quad \text{where } l, j = 1, \ldots, m.$$

By the linearity of the operator $\Phi_i$ and the smoothness of the activation function as in Assumption B.2, it holds that

$$\partial^2_{\theta_j \theta_l} \Phi_i[u_\theta] = \Phi_i \left[ \partial^2_{\theta_j \theta_l} u_\theta \right].$$

For a specific choice, e.g. first parameter is $\theta_j = W_j^1$ and the second is $\theta_l = W_l^0$, it holds that

$$\left| \partial^2_{W_j^1 W_l^0} \Phi_i[u_\theta](x) \right| = \left| \Phi_i \left[ \partial^2_{W_j^1 W_l^0} u_\theta \right] \right| = \left| \frac{1}{\sqrt{m}} \Phi_i[\sigma'(W_l^0 x + b_l^0)x] \mathbf{1}_{l=j} \right| \leq C \frac{1}{\sqrt{m}}, \quad (19)$$

where the last inequality follow from Assumption B.1, Assumption B.2 and the boundedness of the domain $\Omega$.

Since the calculations of (19) are similar for every combination of parameters $W^1, W^0, b^0$, we do not report them here. Furthermore, we notice that the derivatives involving $b^1$ are zeros and hence we obtain that $H_{\Phi_i}$ is composed by 9 blocks ($3 \times 3$ combinations of parameters). Each block is a diagonal matrix, whose elements are bounded by $C \frac{1}{\sqrt{m}}$. By considering that the spectral norm of a diagonal matrix is equal to the maximum of its components, we can bound the spectral norm of each

block by $C\frac{1}{\sqrt{m}}$. Moreover the spectral norm of a matrix can be bounded by the sum of the spectral norm of its blocks, hence:

$$\|H_{\Phi_i}(x)\| \leq 9C\frac{1}{\sqrt{m}} = O(\frac{1}{\sqrt{m}}).$$

$\square$

We can now prove Proposition 3.7.

*Proof.* In the nonlinear case the Hessian of the residuals is

$$(H_r(x))_{j,l} = \partial^2_{\theta_l\theta_j} r_\theta(x) = \partial_{\theta_l}(\nabla R(\Phi[u_\theta](x)))\partial_{\theta_j} u_\theta(x) =$$
$$= \underbrace{\langle \partial_{\theta_l}\Phi[u_\theta](x), \nabla^2 R(\Phi[u_\theta](x))\partial_{\theta_j}\Phi[u_\theta](x)\rangle}_{A_{ij}} + \underbrace{\nabla R(\Phi[u_\theta](x))H_\Phi(x)}_{B_{ij}}.$$

for every collocation point $x \in \Omega$. The matrix $H_\Phi$ is defined in (17). Moreover, Lemma E.1 provides that the spectral norm of $B$ goes to 0 in the infinite-width limit. Moreover, by making use of the inverse triangular inequality, we obtain that for any $x \in \Omega$, it holds

$$\lim_{m\to\infty} \|H_r(x)\| \geq \lim_{m\to\infty} |\|A\| - \|B\|| = \lim_{m\to\infty} \|A\|.$$

According to the definition of spectral norm, we have that

$$\lim_{m\to\infty} \|A\| = \lim_{m\to\infty} \max_{\|z\|_2\leq 1} \|Az\|_2 \geq \lim_{m\to\infty} \|A\bar{z}\|_2,$$

where $\bar{z} := \begin{bmatrix} \frac{1}{\sqrt{m}} & \frac{1}{\sqrt{m}} & \cdots & \frac{1}{\sqrt{m}} \end{bmatrix}$. Let us now focus on the term $\|A\bar{z}\|_2$. By using some standard inequalities and taking advantage of the fact that each entry of $\bar{z}$ is $\frac{1}{\sqrt{m}}$, we obtain that

$$\|Az\|_2 \geq \frac{1}{\sqrt{m}}\|Az\|_1 \geq \frac{1}{\sqrt{m}}\sum_{i=1}^{m}(Az)_i = \frac{1}{m}\sum_{i,j=1}^{m} A_{ij}$$

$$= \frac{1}{m}\sum_{i,j=1}^{m}\langle\partial_{\theta_i}\Phi[u_\theta](x), \nabla^2 R(\Phi[u_\theta](x))\partial_{\theta_j}\Phi[u_\theta](x)\rangle =$$

$$= \left\langle \frac{1}{\sqrt{m}}\sum_{i=1}^{m}\partial_{\theta_i}\Phi[u_\theta](x), \nabla^2 R(\Phi[u_\theta](x))\frac{1}{\sqrt{m}}\sum_{j=1}^{m}\partial_{\theta_j}\Phi[u_\theta](x)\right\rangle$$

Without loss of generality, we can restrict our focus to $\theta_i = W_i^1$ and $\theta_j = W_j^1$, since the spectral norm of a matrix is greater or equal then the norm of its submatrix, and study the term

$$\lim_{m\to\infty}\frac{1}{\sqrt{m}}\sum_{i=1}^{m}\partial_{\theta_i}\Phi[u_\theta](x) = \lim_{m\to\infty}\frac{1}{m}\sum_{i=1}^{m}\Phi[\sigma(W_i^0 \cdot +b_i^0)](x) =$$
$$= \mathbb{E}_{u,v\sim\mathcal{N}(0,1)}[\Phi[\sigma(u\cdot +v)](x)] =: w \tag{20}$$

by the law of large numbers. In particular, $w$ is deterministic. Notice that here we have considered a generic $\theta$ since, according to Lemma C.4, $\partial_{\theta_i}\Phi$ is constant. By combining this result with the previous one, we obtain that

$$\lim_{m\to\infty}\|H_r(x)\| \geq w^T\nabla^2 R(\Phi[u_\theta](x))w \geq \tilde{c}$$

where $\tilde{c}$ is a deterministic constant that does not depend on $m$, but only on the value of $\nabla^2 R$ (which is constant because $R$ is a second-order polynomial) and on the vector $w$ defined in (20). $\square$

## F  Proof of Theorem 4.2

*Proof.* The gradient flow equation in case of Gauss-Newton methods has been defined in (14) for $J(t) \in \mathbb{R}^{n\times p}$ where $t \in [0, T]$. It follows that

$$\begin{bmatrix}\partial_t u_\theta(t) \\ \partial_t r_\theta(t)\end{bmatrix} = \begin{bmatrix}\partial_\theta u_{\theta(t)} \\ \partial_\theta r_{\theta(t)}\end{bmatrix}\partial_t\theta(t) = J(t)\partial_t\theta(t) = -J(t)(J^T(t)J(t))^\dagger J^T\begin{bmatrix}u_{\theta(t)} \\ r_{\theta(t)}\end{bmatrix},$$

where the last equality comes from plugging in (14) into the equation. Now, let us consider the case when $p >> n$, then the singular value decomposition of $J(t)$ is as follows

$$J(t) = U \underbrace{\begin{bmatrix} \tilde{\Sigma}_n & 0_{p-n} \end{bmatrix}}_{\Sigma} V^T,$$

where $U \in \mathbb{R}^{n \times n}, \Sigma \in \mathbb{R}^{n \times p}, V \in \mathbb{R}^{p \times p}$ and $\tilde{\Sigma}_n \in \mathbb{R}^{n \times n}$ is a diagonal matrix with elements given by the square roots of the eigenvalues of the NTK. We drop the dependence on time $t$ of $U, \Sigma$ and $V$ to ease the notation. Let us now study the term

$$
\begin{aligned}
J(t)(J^T(t)J(t))^\dagger J^T(t) &= U\Sigma^T V^T (V\Sigma^T U^T U\Sigma V^T)^\dagger V\Sigma^T U^T \\
&= U\Sigma V^T V (\Sigma^T \Sigma)^\dagger V^T V \Sigma^T U^T \\
&= U\Sigma (\Sigma^T \Sigma)^\dagger \Sigma^T U^T \\
&= U \begin{bmatrix} \tilde{\Sigma}_n & 0_{p-n} \end{bmatrix} \left( \begin{bmatrix} \tilde{\Sigma}_n \\ 0_{p-n} \end{bmatrix} \begin{bmatrix} \tilde{\Sigma}_n & 0_{p-n} \end{bmatrix} \right)^\dagger \begin{bmatrix} \tilde{\Sigma}_n \\ 0_{p-n} \end{bmatrix} U^T \\
&= U \begin{bmatrix} \tilde{\Sigma}_n & 0_{p-n} \end{bmatrix} \begin{bmatrix} \tilde{\Sigma}_n^2 & 0_{p-n} \\ 0_{p-n} & 0_n \end{bmatrix}^\dagger \begin{bmatrix} \tilde{\Sigma}_n \\ 0_{p-n} \end{bmatrix} U^T \\
&= U\tilde{\Sigma}_n (\tilde{\Sigma}_n^2)^\dagger \tilde{\Sigma}_n U^T \\
&= UDU^T
\end{aligned}
$$

where $D$ is obtained from $\tilde{\Sigma}_n$ by replacing the non-zero components with 1. In particular we can rewrite the Gauss-Newton flow as:

$$\begin{bmatrix} \partial_t u_{\theta(t)} \\ \partial_t r_{\theta(t)} \end{bmatrix} = -UDU^T \begin{bmatrix} u_{\theta(t)} \\ r_{\theta(t)} \end{bmatrix}.$$

Notice that it has the same form of the gradient flow in Lemma 3.2 but the Neural Tangent Kernel is replace by a matrix with non-zeroes eigenvalues 1. This can be translated as: second-order optimizers are almost spectrally unbiased. Moreover if $J(t)$ stays full rank during the training, we can obtain the result of convergence regardless of the singular values of $J(t)$, i.e.:

$$\begin{bmatrix} \partial_t u_{\theta(t)} \\ \partial_t r_{\theta(t)} \end{bmatrix} = - \begin{bmatrix} u_{\theta(t)} \\ r_{\theta(t)} \end{bmatrix}.$$

. $\qquad\qquad\qquad\qquad\qquad\qquad\qquad\qquad\qquad\qquad\qquad\qquad\qquad\qquad\qquad\square$

# G  Details about the Numerical Experiments

## G.1  The LM Algorithm

In the following, we provide a more detailed description of the version of the Levenberg-Marquardt algorithm along with its pseudocode and the details of the experiments whom results are shown in Section 5.

The main difference between the Levenberg-Marquardt algorithm and other Quasi-Newton method is that general Quasi-Newton methods are line-search approaches, while LM is a trust region approach. In practice, line search approaches determine a descent direction of the loss function and thereinafter determine a suitable step size in such direction. On the other hand, a trust region method determines an area where the solution lies and computes the optimal step. If this step does not provide enough improvement in the objective function, the search area is reduced and the search is performed once more. We refer to [27] for a thorough description of trust region and line search methods.

In the following part, we drop the dependence on training time as a continuous function and identify $f(t_k) = f_k$ for some discrete time $t_k$. As already mentioned in Section 5, the update step $v_k$ of the LM algorithm is computed follows:

$$v_k = - \left[ J_k^T J_k + \lambda D_k \right]^{-1} \nabla L(\theta_k), \tag{21}$$

where $D_k$ is a diagonal matrix of size $n \times n$. In the classical LM algorithm, this matrix $D_k$ is given by the identity matrix. Another viable alternative recommended in [6] is to use the diagonal of $J_k^T J_k$.

For our model, we choose $D_k$ to be simply the identity matrix, which appears to be more stable when $J_k^T J_k$ is singular.

Another typical modification to the Levenberg-Marquardt algorithm is the introduction of the geodesic acceleration [36].

$$a_k = - \left[ J_k^T J_k + \lambda D_k \right]^{-1} v_k H_r v_k. \tag{22}$$

The goal of the geodesic acceleration is to introduce a component which does consider all the components of the Hessian of the loss when the residuals are not small and when the Hessian of the residuals is not negligible.

Moreover, at every iteration, one has to specify a criterion $C_k$ whose objective is to evaluate the relative improvement of the model parameterized by $\theta_k$ with respect to the update step $v_k$. The criterion depends on the modification of the LM algorithm chosen. For our algorithm we use the same condition as [9] i.e. $C_k < toll$ where $C_k$ is defined as

$$C_k = \frac{L(\theta_k)^2 - L(\theta_k + v_k)^2}{\langle v_k, \lambda_k D_k v_k + \nabla L(\theta_k) \rangle}. \tag{23}$$

We provide in Algorithm 1 the pseudocode of the modified LM algorithm that we chose for our numerical experiments, inspired by the implementation of [9] and modifying it by adding the component of the geodesic acceleration.

---

**Algorithm 1** Modified Levenberg-Marquardt Algorithm

---

**Input:** Maximum region area $\Lambda > 0$, Region Radius $0 < \lambda_0 < \Lambda$, Tollerance $tol \in [0, \frac{1}{4})$, $\alpha \in [0, 1)$
**for** $k = 0, 1, 2, \ldots$ **do**
    Compute $v_k$ as in Equation (21)
    Compute criterion $C_k$ as in Equation (23)
    **while** $C_k < tol$ **do**
        $\lambda = \min(2\lambda, \Lambda)$
        Compute $v_k$ with the new value of $\lambda$
        Compute criterion $C_k$ as in Equation (23)
    **end while**
    $\theta_{k+1} = \theta_k + v_k$
    $\lambda_{k+1} = \max(\frac{1}{3}\lambda, \Lambda^{-1})$
    Compute $a_k$ as in Equation (22)
    **if** $2||a_k|| \leq \alpha ||v_k||$ **then**
        $\theta_{k+1} = \theta_{k+1} + \frac{1}{2} a_k$
    **end if**
**end for**

---

The main focus of the Levenberg-Marquardt method is to decide the size of the trust region. In practice, at every iteration, one wants to find a better solution and afterwards reduce the size of the trust region. When this does not happen, the solution is to enlarge the trust region in order to look for a better solution. In our method we choose to include the region search as part of the inner loop, as for line search approaches. This means that the iteration itself can be slower, but more accurate, which is why we include in the numerical evaluation also the computational time.

### G.2 Poisson Equation

The Poisson equation that we choose for our study is a monodimensional instance of the PDE defined in [39] for $x \in \Omega = [0, 1]$ and we try to find the solution $u : \Omega \to \mathbb{R}$. In particular, we want to solve the following equation:

$$\begin{aligned} \partial_x^2 u &= f(x), \quad x \in \Omega, \\ u(0) &= u(1) = 0. \end{aligned} \tag{24}$$

As in [39], the function $f$ is constructed in such a way that the exact solution of Equation (24) is given by:

$$u(x) = \sin(2\pi x) + \frac{1}{10} \sin(50\pi x).$$

This approach is done to evaluate the behavior of PINNs when the target solution presents a high frequency and a low frequency component. We then train the PINN model by sampling $N_r = 10^3$ points in $\Omega$ with latin hypercube sampling.

### G.3   Wave Equation

We opt to solve the wave equation below for each $(x, \tau) \in \Omega = [0, 1]^2$ and aim to find the solution $u : \Omega \to \mathbb{R}$. In particular, we aim to solve the following equation:

$$
\begin{aligned}
\partial_\tau^2 u &= -C^2 \partial_x^2 u, & (x, \tau) &\in \Omega, \\
u(x, 0) &= \sin(\pi x) + \frac{1}{2}\sin(4\pi x), & x &\in [0, 1], \\
\partial_\tau u(x, 0) &= 0 & x &\in [0, 1], \\
u(0, \tau) &= u(1, \tau) = 0, & \tau &\in [0, 1].
\end{aligned}
\tag{25}
$$

With $C$ being equal to 2 for our case. It is straightforward to obtain the correct solution of this equation through Fourier transform. In particular, the exact solution of Equation (25) is given by:

$$
u(x, \tau) = \sin(\pi x)\cos(2\pi\tau) + \frac{1}{2}\sin(4\pi x)\cos(8\pi\tau).
$$

We then train a PINN by sampling $N_r = 10^4$ training points in $\Omega$ for the PDE residuals with latin hypercube sampling, and $N_b = 3 \cdot 10^3$ points for training the model against the correct solution at $\partial\Omega$.

### G.4   Burgers' Equation

Burgers' equation is a 1D version of Navier-Stokes equations. Its solution at high times present a discontinuity, which makes it challenging for spectrally biased architectures. The specific instance chosen in our numerics for Burgers' equation is the same as in [30]. In particular, we refer to the exact same data provided by the authors. In particular, given $(x, \tau) \in \Omega = [-1, 1] \times [0, 1]$, we solve for $u : \Omega \to \mathbb{R}$ the following equation:

$$
\begin{aligned}
\partial_\tau u + u\partial_x u - \nu \partial_x^2 u &= 0, & (x, \tau) &\in \Omega, \\
u(x, 0) &= -\sin(\pi x), & x &\in [-1, 1], \\
u(-1, \tau) &= u(1, \tau) = 0, & \tau &\in [0, 1],
\end{aligned}
\tag{26}
$$

with the diffusivity $\nu$ being equal to $\frac{0.01}{\pi}$ for this specific instance. The correct solution is provided publicly by the authors of [30].

Training is performed with $N_r = 10^4$ collocation points for training the PDE residuals, sampled with latin hypercube sampling, and $N_b = 3 \cdot 10^3$ points for training the boundary and initial condition in $\partial\Omega$.

### G.5   Navier-Stokes Equation

The most interesting scenario taken in consideration for our experiments is that of Navier-Stokes equations. In particular, we aim to solve the fluid flow in the wake of a cylinder in 2D tackled in [17]<. In particular, we have $(x, y, t) \in \Omega = [2.5, 7.5] \times [-2.5, 2.5] \times [0, 16]$ and we wish to find $\vec{u} : \Omega \to \mathbb{R}^3$ which is defined as $\vec{u}(x, y, t) = [u(x, y, t), v(x, y, t), p(x, y, t)]^T$. In particular $u$ and $v$ are respectively the horizontal and vertical components of the fluid velocity and $p$ is the pressure at a point. Navier-Stokes equations are then expressed in vectorizer form as follows:

$$
\begin{aligned}
\partial_\tau u + u\partial_x u + v\partial_y u - \frac{1}{Re}\left(\partial_x^2 u + \partial_y^2 u\right) + \partial_x p &= 0, & (x, y, \tau) &\in \Omega, \\
\partial_\tau v + u\partial_x v + v\partial_y v - \frac{1}{Re}\left(\partial_x^2 v + \partial_y^2 v\right) + \partial_y p &= 0, & (x, y, \tau) &\in \Omega, \\
\partial_x u + \partial_y v &= 0, & (x, y, \tau) &\in \Omega, \\
u(x, y, 0) &= g_{u_0}(x, y), & (x, y) &\in [2.5, 7.5] \times [-2.5, 2.5], \\
v(x, y, 0) &= g_{v_0}(x, y), & (x, y) &\in [2.5, 7.5] \times [-2.5, 2.5], \\
u(2.5, y, \tau) &= 1, & (y, \tau) &\in [-2.5, 2.5] \times [0, 16], \\
v(2.5, y, \tau) &= 0, & (y, \tau) &\in [-2.5, 2.5] \times [0, 16],
\end{aligned}
\tag{27}
$$

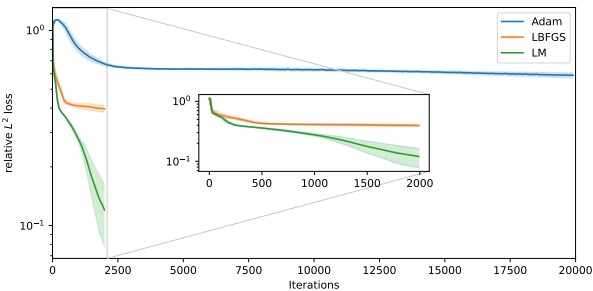

Figure 5: Mean and standard deviation of the relative $L^2$ loss on the test set on the Wave equation for Adam, L-BFGS and LM optimizer over iterations (repetition over 10 independent runs).

where $Re$ represents the Reynolds' number, which is an adimensional quantity defined by the problem and is set to 100 for our case. The initial conditions $(g_{u_0}, g_{v_0})$ can be found in the repository published by the authors of [32], as well as the correct solution. The conditions at $x = -2.5$ represents the fluid velocity imposed at the inlet, and further conditions are given by the presence of a cylinder centered in $(x, y) = (0, 0)$ with radius 0.25. Furthermore, an additional condition appears at the borders, namely where $y = \pm 2.5$, where the no-slip condition can be chosen ($u = v = 0$) or the correct solution can be given as boundary condition. Since the simulation provided in [32] refers to a free-flow stream, we use the correct solution at the boundaries.

To train our PINNs, we use $N_r = 5 \cdot 10^5$ collocation points for training the PDE residuals, sampled with latin hypercube sampling, and $N_b = 2 \cdot 10^4$ points for training the boundary and initial condition in $\partial\Omega$. Morever, at every iteration, we minimize the loss on random batches of the training data, respectively $10^4$ points for the residuals and $5 \cdot 10^3$ for boundary and initial condition.

## H    Further Numerical Experiments

In this Appendix we present some additional numerical experiments. Notice that as a performance measure we utilize the $L^2$ relative loss, defined as follows

$$\sum_{i=1}^{N} \frac{|u(x_i) - \hat{u}(x_i)|}{|u(x_i)|}, \tag{28}$$

where $u$ is the exact solution and $\hat{u}$ the approximated one.

In Figure 5, we showcase the relative $L^2$ loss obtained on the test set during training on the Wave equation with the aforementioned optimizers. While Adam and L-BFGS get stuck relatively fast in a local minima, the LM algorithm is able to decrease the loss consistently, despite the complexity of the problem. The poor performance of L-BFGS can be motivated by two factors. On one hand, the Hessian computed during BFGS iterations is merely an approximation of the true Hessian; on the other hand, convergence to the true solution is heavily hindered since the initial guess is typically not close to the correct one.

In Figure 6 and Figure 7, it is possible to notice the effect of the spectral bias: the PINN trained with Adam can capture only the lower frequency components of the true solution, while the model trained with LM performs better as the spectral bias is alleviated in accordance with Theorem 4.2. It is worth noticing that the same holds even when introducing the loss balancing suggested in [40]: its performance is showed in Figure 7.

Finally, in Figure 8, we show that by employing the LM optimizer, it is possible to obtain a reasonable solution even for a PDE as complex as Navier-Stokes with relatively small architectures. Notice that the scale in the two plots are different.



Figure 6: Experiments on the Wave equation. Left: Prediction of the parametrized solution of a PINN trained with Adam (Left) and LM (Center) alongside with the true solution (Right).

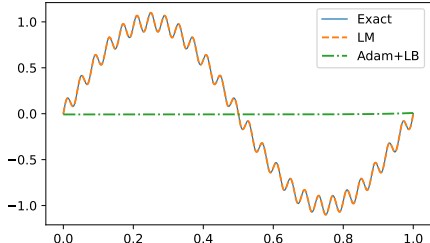

Figure 7: Experiments on the prediction of the solution of Poisson equation with LM and Adam (with loss balancing), both compared with the exact solution.

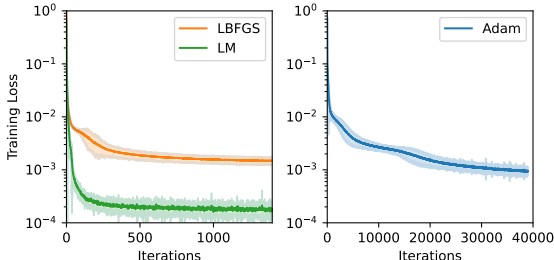

Figure 8: Mean and standard deviation of the training loss over the iterations for Adam, LBFGS and LM on Navier-Stokes equation (for 10 independent runs).

