# OpenReview forum: "The Challenges of the Nonlinear Regime for Physics-Informed Neural Networks"
_NeurIPS.cc/2024/Conference — NeurIPS 2024 poster_

### Official Review · Reviewer_XYBU · 2024-07-04

**Soundness:** 3
**Presentation:** 4
**Contribution:** 3
**Rating:** 6
**Confidence:** 3

**Summary:**

The authors analyze the NTK perspective for PINNs for non-linear PDEs. Previous considerations derived for linear PDEs fall short for non-linear ones and the authors attribute this difference to the non-vanishing Hessian term. Therefore, they suggest to use second order methods and show to be able to converge faster in one experiment. Second order method are shown to be useful also for linear PDEs as they alleviate the spectral bias of the NTK.

**Strengths:**

- the paper is well written and flows very nicely
- I found the analysis about the NTK in non-linear regime elegant and the experiments clearly support the findings

**Weaknesses:**

- the different behaviour of the NTK for non-linear PDEs is not very surprising
- it is also not very surprising that second order method can work better, but in practice they come with significant shortcomings. Even though the method is shown to be faster in one experiment, no rate is derived so it might not be true in general.

However, the applicability of Theorem 4.2 is just sketched (lines 218-222)

**Questions:**

- I would like the authors to elaborate on the applicability of the second order method. In particular, I found the explanation in lines 218-222 a bit hand-wavy. It seems like the hypothesis of Theorem 4.2 (e.g. J(t) being full-rank) are very hard to check in practice
- No guidance or intuition is provided for practitioners on when it might be convenient to use the second order method, given that the method is expensive (the authors only mention general well-known considerations for second order methods in the limitations paragraph)

**Limitations:**

Limitations are discussed. However, as mentioned in the question section, I think the authors should discuss in more details the scalability of the proposed second order approach and on when it might be convenient to use it.

---

> ### Author Rebuttal · Authors · 2024-08-06
>
> We sincerely thank the reviewer for the kind words regarding the structure and analysis in our paper. While we agree that some of the results might be expected, most of the existing literature focuses on the linear regime and only conjectures what might happen in the nonlinear case. To our knowledge, our paper is the first to characterize the NTK dynamics in the nonlinear case. This characterization is crucial for understanding that the poor behavior of the Hessian can lead to slow and unreliable training of PINNs.
>
> - **[Q1]** We understand the reviewer's concerns regarding the applicability of Theorem 4.2. However, proving the full-rankness of $J$ in the nonlinear case is an open problem that has been widely discussed and remains unresolved. Our contribution is to highlight that the NTK is stochastic in the limit, and for this reason, if one wants to check the full-rankness of $J$, probabilistic methods need to be used. We leave this task to future works. More generally, we do not assert that $J$ is necessarily always full-rank. The remark in lines 218-222 is meant to highlight that the well-conditioning of $J$ is not crucial for a better training when using a second-order method. Indeed, one of the advantages of using second-order methods is that the diagonal of the matrix $D$, even when $J$ is ill-conditioned, only contains ones and zeros.
>     While a first-order method would yield,  in addition to zeros, values that might be extremely small (i.e. on the order of $10^{-14}$ in figure 2(b)). Regardless, this assumption is weaker than the one needed for theoretical convergence guarantees of first-order methods.
>
>  - **[Q2]** We thank the reviewer for the interesting question, and we believe that adding a paragraph on this would improve the quality of our paper. In an improved version, we plan to add an intuition on when to use second-order methods. In practice, it may be convenient to use second-order methods whenever there are high frequencies in the solution, when the application requires high accuracy, and when the PDE we aim to solve is nonlinear. One of the goals of our paper is to emphasize that, despite their shortcomings, second-order method are in general a natural choice in the field of PDE solution.

---

> > ### Comment · Reviewer_XYBU · 2024-08-14
> >
> > I would like to thank the authors for their clarifications

---

### Official Review · Reviewer_ocfJ · 2024-07-08

**Soundness:** 3
**Presentation:** 4
**Contribution:** 3
**Rating:** 7
**Confidence:** 5

**Summary:**

This paper studies the training dynamics of PINNs, especially for the nonlinear PDEs. The authors find that the previous recognized NTK viewpoint is not applicable to nonlinear PDEs, although it holds for linear PDEs. Therefore, the global convergence of gradient descent on nonlinear PDEs may not be guaranteed. Moreover, the imbalance of singular values of Gram matrices, which also occurs in linear PDEs, results in slow convergence. To address this issue of spectral bias, the paper suggests using second-order methods for parameters update.  Experimental results show that LM Newton’s method can achieve lower training loss compared to Adam and L-BFGS.

**Strengths:**

The paper is well-written and provides both theoretical and empirical analyses. The failure of NTK approach on nonlinear PDEs has not been highlighted in previous studies of the global convergence of training PINNs with gradient descent. The second-order method is essential in smooth problems and deep learning training. The authors highlight the effectiveness of Newton’s method in balancing singular values of the training dynamics.

**Weaknesses:**

(1) the failure of NTK is similarly investigated in some previous works (e.g., https://www.sciencedirect.com/science/article/pii/S016727892300341X), where they found that the Gram matrix does not consistently converge in some cases. Therefore, the observation seems to be not novel. They also pointed out that NTK may also work for some nonlinear PDEs.
(2) the second-order methods for modifying the singular values of Gram matrix of training dynamics are also not new.
(3) the Hessian inverse is quite expensive in practice.
(4) regularizing the Hessian may also be intractable in practice, for high dimensional problems, e.g., training PINNs.

**Questions:**

I have the following questions:
(1) In the previous work (e.g., https://www.sciencedirect.com/science/article/pii/S016727892300341X), it was found that NTK approach fails in solving some PDEs. Besides linear PDEs, it is still possible that NTK works for some nonlinear PDEs. However, in your paper, you exclude the special situation. It seems to be more restrictive than the published work.
(2) Although the second-order methods enjoy very good theoretical properties, deep learning community typically prefers first-order methods due to their computational efficiency. In practice, Newton’s method is more computational expensive even with some inexact technique (e.g., Krylob subspace, conjugate gradient, and LBFGS). However, your theorem of global convergence (Theorem 4.2) does not apply to these inexact methods. Based on your result (which I believe built upon some previous works), can you extend your findings to more practical inexact Newton’s method (e.g., replacing Hessian inverse by Krylov subspace method or Quasi-Newton’s method of BFGS or LBFGS)?
(3) In your experimental results, the LM method performs significantly better than LBFGS. This is surprising, as LBFGS or BFGS asymptotically approaches the exact Hessian under certain conditions. Moreover, in case where the totally number of grid points are large and the batch is small in a stochastic setting, inexact and Quasi Newtons method (e.g., Hessian averaging methods including BFGS and LBFGS) should hold the asymptotic convergence. Therefore, intuitively, I would expect LBFGS to perform at least comparably with your LM method. Am I correct? Can you explain why LBFGS performs poorly according to your results, although it can approximate Hessian and extract Hessian information?

**Limitations:**

Yes

---

> ### Author Rebuttal · Authors · 2024-08-06
>
> We would like to extend our gratitude to the reviewer for their thoughtful feedback and for bringing this reference to our attention. Now, we will proceed to address the specific questions and concerns raised by the reviewer.
>
> - **Q1** In the paper referenced by the reviewer, the convergence of the Gram matrix at initialization (Theorem 2.2) holds in the non-homogeneous case only for $s\geq1$.  The authors state that for $s=\frac{1}{2}$, it is impossible to guarantee universal convergence. This is why our results may seem more restrictive at first glance, but this is due to our focus on the typical NTK scaling $N^{-\frac{1}{2}}$, which allows us to obtain meaningful characterization of the training dynamics using kernel analysis. Furthermore, we explicitly characterize the law of the NTK's limit. In addition to being applicable to any nonlinear PDE, this result enables the leveraging of random matrix theory tools to prove its positive definiteness at initialization, which is a longstanding conjecture in the research community. Hence, we would like to highlight that our results do not contradict those in the referenced paper, even if there seem to be inconsistencies that we are now going to address.
>
>     - Regarding the constancy of the Gram matrix during training (Theorem 2.3), we have noticed a problem in the proof of Lemma C.4. In equations 63-68, the authors establish a bound for a single parameter as $ O(1/N^{2s-1/2})$. However, there might be an inconsistency in equation 69, where the same bound is applied to the full vector, which has $O(N)$ components. This discrepancy implies that the bound should be $O(\sqrt{N} \cdot 1/N^{2s-1/2}) = O(1/N^{2s-1})$. Consequently, their theorem would hold for $ s > 1/2 $ instead of $s > 1/4$. Hence, this theorem and our Proposition 3.5 do not contradict each other, since we study $s=1/2$.
>
>     - Finally, there seem to be differences regarding the numerical experiments in the reference. However, we noticed that the equations studied in Figures 3 and 5 have a consistent linear part, while our PDE is fully nonlinear. The variation of $K(t)$  during training decreases as $N$ grows because the variation of its linear components decreases, according to our theory. However, the nonlinear components still evolve, and for even larger $N$ (we use a number of neurons several orders of magnitude greater), we observe the plateauing of the plots at a value strictly greater than 0 (note that in their Figure 5, the y-axis starts from 10).
>
> - **Q2 & Q3** To extend our findings to a more practical inexact Newton's method, we can indeed utilize the approach you suggested in Q3, specifically the asymptotic convergence properties of LBFGS and other methods. While this approach is theoretically sound, we must note the following. The theoretical guarantees on the speed of convergence of quasi-Newton methods to exact Newton methods, i.e. their matrix approximation ability, depend on the minimum eigenvalue of the Hessian ([1], Theorem 6). However, as discussed in our paper, the Hessian in PINNs is generally very poorly conditioned. Consequently, quasi-Newton methods may require a practically infinite number of training steps to converge to the true Hessian's inverse and, therefore, to start training higher modes. The aforementioned reasons would explain the intermediate performance of LBFGS, which lies between that of first-order and exact second-order methods.
>
> **[1]** Lin, D., Ye, H., and Zhang, Z. (2022). Explicit convergence rates of greedy and random quasi-Newton methods. Journal of Machine Learning Research.

---

> > ### Comment · Reviewer_ocfJ · 2024-08-09
> > **Answer to the rebuttal**
> >
> > Thank you for the clarification. My concerns are well addressed. I would like to raise my score. To further improve the quality of the paper and make clear claim, I hope authors can include the above related discussions (e.g., the convergence of gram matrices holds for nonlinear PDEs when s>1/2, and the extension (although may fail in practice) to other quasi-newton methods).

---

> > > ### Author Response · Authors · 2024-08-13
> > >
> > > We are glad that our reply effectively addressed the reviewer's concerns and we deeply appreciate the increase in score. It will be our pleasure to include in our paper the discussion above.

---

### Official Review · Reviewer_4Qrf · 2024-07-09

**Soundness:** 3
**Presentation:** 3
**Contribution:** 4
**Rating:** 7
**Confidence:** 3

**Summary:**

The paper studies the NTK of NNs trained on non-linear PDEs, showing that they exhibit different behaviours compared to standard analysis of NTKs. The paper then discusses the issue of spectral bias that arises from first-order methods, showing that they can be alleviated by the use of second-order methods.

**Strengths:**

- The paper presents an interesting analysis of a common tool in NNs and PINNs, and presents an explanation why second-order methods (which are already used in PINNs to some extent) works better than first-order methods.
- The paper provides both theoretical and empirical justification for the various claims, and is well-organised in that manner.

**Weaknesses:**

- The sections could be a bit more coherent. For example, the paper brings up the properties of the NTK in the nonlinear PDE case, but then provides less link of these properties of how it affects the convergence in terms of the spectral biases. The LM algorithm is also brought up as a second-order optimisation method, however it may warrant more description as to why it is introduced or how it differs from existing second-order methods such as LBFGS.
- Explicit mention of LM algorithm's runtime could be mentioned for completeness.

**Questions:**

- Already in PINNs, there are many works that uses NTKs in loss function scaling [1], collocation point selection [2], analysis of PINN architectures [3], and more. How would the insights in the paper be able to address the points raised in these papers, and how would they affect these proposed methods?
- Is Theorem 4.2 general enough to be applied to regular NNs as well? How does the result compare to existing theoretical works on second-order methods in NNs or general optimisation problems?

[1] Wang et al. When and why PINNs fail to train: A neural tangent kernel perspective.

[2] Lau et al. PINNACLE: PINN Adaptive ColLocation and Experimental points selection.

[3] Wang et. al. On the eigenvector bias of Fourier feature networks: From regression to solving multi-scale PDEs with physics-informed neural networks.

**Limitations:**

Limitations suggested are of the LM algorithm which is adequate.

---

> ### Author Rebuttal · Authors · 2024-08-06
>
> We would like to thank the reviewer for the positive feedback and the constructive comments on our paper. In particular, we acknowledge the second weakness highlighted by the reviewer. We plan to include this information in an improved version of our paper. At present, the reviewer can get a qualitative estimate of the runtime of LM with comparison to that of Adam and L-BFGS by referring to Figure 4(a). Now, let us address the reviewer's questions in the order they were asked:
> - **Q1** We would like to emphasize that our intention is not to suggest that our method should substitute other existing approaches, such as the ones mentioned by the reviewer. On the contrary, we believe that combining second-order methods with the enhancements to PINNs mentioned above can yield highly competitive results. This potent combination is demonstrated in Figure 3(a), where we used the LM algorithm together with the method proposed in [3] (as per the reviewer's nomenclature) resulting in fast and accurate convergence on a strongly spectrally biased PDE. On the right side of the same figure, we compare and combine our method with another PINN enhancement, namely curriculum training. Thus, we believe that the LM algorithm, in conjunction with the collocation point selection presented in [2], could also result in positive improvements, especially for engineering use cases where traditional solver often relies on adaptive meshes. We appreciate the reviewer for highlighting this work. Regarding reference [1], the issue of unbalanced loss components is addressed through loss scaling. However, in our work, this issue is implicitly tackled with second-order methods according to the result presented in Theorem 4.2. Indeed, the presence of ones in the convergence matrix $D$ implies, in particular, that the various loss components are balanced during training, producing excellent results when compared to loss scaling. We already have a comparison with [1] in the Appendix, Figure 7, where we referred to the method in [1] as "loss balancing" instead of "loss scaling."
>
> - **Q2** We believe that Theorem 4.2 can also be applied to regular NNs, as there are no specific assumptions on the loss function other than the requirement that the Jacobian $J$ has to be full-rank. However, Theorem 4.2 specifically addresses the issue of spectral bias in PINNs. This issue is a well-known problem for PINNs (even for linear PDEs), but we are not aware of any correlation between spectral bias and worse performance in regular NNs' tasks. Nevertheless, we would like to highlight that there are several papers, such as [4, 5, 6], where general second-order methods have been successfully employed to train regular NNs. Hence, it might be beneficial to extend our analysis to these cases.
>
> **[4]** Z. Yao, A. Gholami, K. Keutzer and M. W. Mahoney, "PyHessian: Neural Networks Through the Lens of the Hessian," 2020 IEEE International Conference on Big Data.
>
> **[5]** Liu, G. H., Chen, T., and Theodorou, E. (2021). Second-order neural ode optimizer. Advances in Neural Information Processing Systems.
>
> **[6]** Vinyals, O., and Povey, D. (2012). Krylov subspace descent for deep learning. In Artificial intelligence and statistics, PMLR.

---

### Official Review · Reviewer_vTvV · 2024-07-13

**Soundness:** 4
**Presentation:** 3
**Contribution:** 4
**Rating:** 7
**Confidence:** 4

**Summary:**

In this paper, the theory of the Neural Tangent Kernel(NTK) in the case of solving nonlinear partial differential equations using PINNs is investigated in detail. In particular, it is shown that typical results of the NTK framework do not hold when the simple gradient descent method is employed due to the worse behavior of the Hessian matrix compared to the linear cases. In contrast, when second-order methods are employed, it is theoretically proven that the training of the neural networks is efficient.

**Strengths:**

This paper theoretically investigates the behaviors of learning dynamics for PINNs, which are known to be difficult to train. This paper provides a theoretical guarantee of the effectiveness of second-order optimization methods for training PINNs for nonlinear partial differential equations. Im my opinion, this is a significant result, which may lead to applications of PINNs to practical problems that have been inapplicable due to the training difficulties.

**Weaknesses:**

My concern about this paper is in the increase of the computational complexity of second-order methods; however, this concern has already been discussed by the authors in the paper.

**Questions:**

In the numerical experiments, it seems that not so large neural networks are employed. Is it expected that neural networks of this size behave like the results of the theory?

**Limitations:**

There seems to be no problem.

---

> ### Author Rebuttal · Authors · 2024-08-06
>
> First and foremost, we would like to express our sincere gratitude to the reviewer for the encouraging feedback and thoughtful comments on our paper.
>
> Regarding the reviewer's question, you are correct in noting that part of our theoretical framework is developed at the NTK level, i.e., for infinitely wide neural networks. However, our result is "negative" in the sense that, even in this idealized scenario, a first-order method might fail to perform effectively when using PINNs to solve nonlinear PDEs. This, as you mentioned, is due to the worse behavior of the Hessian matrix. Consequently, in the more practical scenario of a network with finite width, we cannot guarantee that PINNs trained with first-order methods can accurately solve any nonlinear PDE.
>
> Moreover, we would like to emphasize that Theorem 4.2, which addresses the convergence of second-order methods, is applicable even for networks with finite width. Therefore, while the NTK model serves as motivation to employ second-order methods, the significant findings regarding spectral bias and convergence do not rely on this idealized model.

---

> > ### Comment · Reviewer_vTvV · 2024-08-13
> >
> > I appreciate the authors for the detailed reply. Because I have already given a high score, I will keep my score.

---

> > > ### Author Response · Authors · 2024-08-13
> > >
> > > We are pleased that our response met the reviewer's expectations and feedback.

---

### Decision · Program_Chairs · 2024-09-25

**Decision:**

Accept (poster)

**Comment:**

The manuscript discusses the NTK regime of PINNs trained on nonlinear PDEs and shows that several results from linear PDEs cannot be shown to carry over to this setting. Specifically, the NTK is not constant during training and the Hessian does not vanish, explaining the superior performance of second-order optimization methods.

All reviewers agree that the topic is relevant and that the presentation of the paper is excellent. The experimental evidence supports the presented theory. Despite some reviewers acknowledging that NTK analyses of PINNs and insights into the performance of second-order methods for PINN training are not new, all reviewers agree that the paper should be accepted.